# Fictitious Synthetic Data Can Improve LLM Factuality via Prerequisite Learning

**Yujian Liu**
UC Santa Barbara
`yujianliu@ucsb.edu`

**Shiyu Chang**
UC Santa Barbara
`chang87@ucsb.edu`

**Tommi Jaakkola**
MIT CSAIL
`tommi@csail.mit.edu`

**Yang Zhang**
MIT-IBM Watson AI Lab
`yang.zhang2@ibm.com`

## Abstract

Recent studies have identified one aggravating factor of LLM hallucinations as the knowledge inconsistency between pre-training and fine-tuning, where unfamiliar fine-tuning data mislead the LLM to fabricate plausible but wrong outputs. In this paper, we propose a novel fine-tuning strategy called PREREQ-TUNE to address this knowledge inconsistency and reduce hallucinations. Fundamentally, PREREQ-TUNE disentangles the learning of skills and knowledge, so the model learns only the task skills without being impacted by the knowledge inconsistency. To achieve this, PREREQ-TUNE introduces an additional prerequisite learning stage to learn the necessary knowledge for SFT, allowing subsequent SFT to focus only on task skills. PREREQ-TUNE can also be combined with fictitious synthetic data to enhance the grounding of LLM outputs to their internal knowledge. Experiments show that PREREQ-TUNE outperforms existing baselines in improving LLM's factuality across short QA and long-form generation tasks. It also opens new possibilities for knowledge-controlled generation in LLMs. Our code is available at `https://github.com/UCSB-NLP-Chang/Prereq_tune.git`.

## 1 Introduction

Hallucination of large language models (LLMs) refers to the phenomenon where LLMs' outputs look plausible but diverge from real-world facts. It has become a major concern of LLMs, seriously undermining their reliability and trustworthiness (Huang et al., 2023; Ji et al., 2023).

Recent research has unveiled one aggravating factor of LLM hallucination, which is the *knowledge inconsistency* between the pre-training and tuning (*e.g.*, instruction- or fine-tuning) stages (Gekhman et al., 2024; Kang et al., 2024; Lin et al., 2024). More specifically, if the tuning stage involves training examples that require knowledge that an LLM has not seen during pre-training, then the LLM would be misled to fabricate plausible but wrong answers to unfamiliar questions (Schulman, 2023; Gao, 2021; Goldberg, 2023). For example, consider fine-tuning a model for a question answering (QA) task with the example *'When was John Estes born?'* and assume that the LLM has never learned about *John Estes* during pre-training. However, since the LLM is still trained to produce the correct answer, *'1987'*, it is consequently encouraged to respond with a random legitimate year whenever it is asked about the birth year of any unknown person, thus giving rise to hallucination.

These findings highlight an important but previously understudied consideration of LLM training, which is the *disentanglement between knowledge and skill*. Specifically, the abovementioned knowledge inconsistency issue indicates that knowledge can interfere with the learning of skills during fine-tuning. If a piece of knowledge required in a fine-tuning example is non-existent in the LLM, training on this example will not teach the LLM the desired skills (*e.g., answering questions based on its internal knowledge); instead, the training will incentivize the LLM to fabricate non-existent facts that look plausible. Thus, only by disentangling knowledge from skill during the tuning stage can we remove such interference.

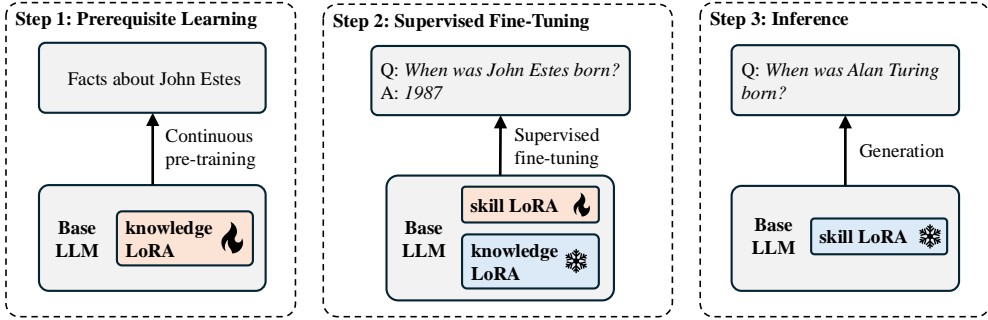

Figure 1: Overview of the proposed PREREQ-TUNE strategy.

An ideal powerful disentangling mechanism should achieve maximal robustness against the knowledge inconsistency, to the extent that even *fictitious synthetic data*, which has zero knowledge overlap with pre-training, would still warrant successful fine-tuning without encouraging hallucination. Unfortunately, this may sound too good to be true, because such an effective disentanglement strategy has yet to be developed for LLM tuning.

In this paper, we propose PREREQ-TUNE, an innovative LLM fine-tuning strategy that explicitly resolves the disentanglement challenge, and therefore can effectively reduce LLM hallucinations. As shown in Figure 1, PREREQ-TUNE consists of two stages, an innovative *prerequisite learning* stage and a supervised fine-tuning (SFT) stage. In the prerequisite learning stage, we train a LoRA (Hu et al., 2022), called the *knowledge LoRA*, that learns all the necessary knowledge required for SFT. In the SFT stage, the knowledge LoRA is frozen, and a new LoRA, called the *skill LoRA*, is imposed on top of the knowledge LoRA, and is trained to perform the SFT task. Since the knowledge LoRA already eliminates the knowledge mismatch issue, the skill LoRA can focus on learning the skills without inducing hallucinations.

With the prerequisite learning as a disentanglement mechanism, PREREQ-TUNE can turn fictitious synthetic data, which is otherwise detrimental to LLM factuality, into a powerful weapon to further reduce LLM hallucinations. In particular, using fictitious synthetic data, we can create multiple knowledge LoRAs that contain different versions of knowledge about the same fictitious entity, and then teach the skill LoRA to produce different answers based on which knowledge LoRA is in use. In this way, we can force the skill LoRA to ground the LLM's answers to its internal knowledge, thus reducing hallucinations. Moreover, unlike real data which are expensive to obtain and label, fictitious synthetic data can be cheaply scaled up, which would further enhance the LLM's factuality thanks to PREREQ-TUNE, informing a new real-data-efficient fine-tuning paradigm.

Our experiments reveal that PREREQ-TUNE can teach an LLM to not only ground its answer to the knowledge LoRAs, but also, more surprisingly, generalize the grounding to the LLM original pre-trained knowledge when the knowledge LoRA is removed. As a result, PREREQ-TUNE can significantly outperform the existing state-of-the-art hallucination reduction algorithms in improving LLM's factuality across short QA and long-form generation tasks. In addition, PREREQ-TUNE enables a more modular design of LLM, with plug-and-play knowledge modules that control the knowledge access and a skill module that works generically with any knowledge sources, which opens up many new possibilities far beyond hallucination reduction, such as novel retrieval augmented generation (RAG) paradigms, privacy protection, *etc*.

## 2 RELATED WORKS

**Reducing LLM hallucinations.** Numerous studies have sought to detect and mitigate hallucinations in LLMs (Weng, 2024; Li et al., 2023a; Chern et al., 2023; Azaria & Mitchell, 2023; Manakul et al., 2023; Chen et al., 2024; Hou et al., 2024). A common approach to reduce hallucinations is leveraging supporting evidence, either retrieved from external knowledge sources or generated by LLMs. Such evidence is often provided as input to help LLMs generate more factual responses (Shuster et al., 2021; Nakano et al., 2022; Menick et al., 2022; Sun et al., 2023b; Asai et al., 2024; Feng et al., 2024); or they can assist in detecting incorrect statements and guide LLMs to post-edit

their own generations (Gao et al., 2023; Dhuliawala et al., 2024; Lei et al., 2023; Mishra et al., 2024). Beyond using additional evidence, several works have proposed decoding algorithms to improve LLM factuality during inference (Lee et al., 2022; Chuang et al., 2024; Li et al., 2023b). More recently, some studies have also explored answer abstention when LLMs encounter unfamiliar questions (Zhang et al., 2024; Yadkori et al., 2024; Yang et al., 2023; Cheng et al., 2024). Our work aligns with recent efforts that fine-tune LLMs so that they inherently generate less hallucinated contents (Tian et al., 2024; Lin et al., 2024; Kang et al., 2024; Ghosal et al., 2024). Unlike existing methods, we focus on the disentanglement of skills and knowledge and the groundedness of LLMs' outputs, which leads to superior performance in hallucination reduction.

**Fine-tuning LLMs with synthetic data.** Synthetic data has shown great potential in fine-tuning LLMs, offering scalability by automatically creating instruction-response pairs with minimal human supervision (Wang et al., 2023; Li et al., 2024; Gupta et al., 2024; Yu et al., 2024; Haluptzok et al., 2023). Additionally, the generation process can be designed to control the training data, allowing models to be trained for specific skills or aligned with particular human values (Sudalairaj et al., 2024; Sun et al., 2023a; 2024; Kaur et al., 2024). However, most works do not consider factuality in fine-tuning. The work closest to ours is Jones et al. (2024), which fine-tunes LLMs on synthetic tasks to reduce hallucinations. However, they focus on improving consistency with evidence provided in the context, whereas we improve LLMs' inherent factuality without provided evidence.

## 3 METHODOLOGY

### 3.1 PROBLEM FORMULATION

Given a pre-trained LLM (without instruction-tuning), parameterized by $\theta_0$, a downstream task $\mathcal{T}$, and its corresponding dataset $\mathcal{D}_{\mathcal{T}}$, our task is to fine-tune the LLM to the downstream task. For concreteness, we will base our illustration of the algorithm on a specific task, *biography generation*. Generalization to other tasks is discussed in Section 3.6. In this task, $\mathcal{D}_{\mathcal{T}}$ would typically contain requests to generate a biography of a certain person, *e.g., 'Generate a biography for John Estes'*, as well as the corresponding human-written biographies.

We will focus on two aspects embedded in $\mathcal{D}_{\mathcal{T}}$ – the knowledge and skill. The knowledge refers to all factual information involved in $\mathcal{D}_{\mathcal{T}}$, and the skill describes the LLM's ability to output the desired format to different user queries, *e.g.,* the ability to answer a question or write biographies in the desired formats. It is worth mentioning that the terms we use here have also been discussed in existing literature. In particular, Zhou et al. (2023) uses the term 'style and format' to describe the LLM's ability to output the desired format, which corresponds to the 'skill' in our paper.

The challenge we aim to resolve is that $\mathcal{D}_{\mathcal{T}}$ may require knowledge that the pre-trained LLM $\theta_0$ does not know, which would encourage LLM hallucination. Specifically, although $\mathcal{D}_{\mathcal{T}}$ involves both knowledge and skill, the difficulty of learning the two is different, and evidence has shown that knowledge and skills are acquired at different stages of the fine-tuning process – skills are generally acquired earlier than knowledge (Gekhman et al., 2024). It is believed that such a gap (in difficulty and learning stages) between knowledge and skill acquisition is a likely cause of hallucination – since skills are learned before the new knowledge is absorbed, LLMs are misled into fabricating non-existent facts whenever they see unfamiliar questions, as a shortcut to minimize fine-tuning loss (Gekhman et al., 2024; Kang et al., 2024; Lin et al., 2024). Our goal is to design a fine-tuning strategy that can learn the skill, *e.g.,* writing biographies, without being impacted by the inconsistency between the knowledge involved in $\mathcal{D}_{\mathcal{T}}$ and the pre-trained knowledge.

### 3.2 BASIC PREREQ-TUNE STRATEGY

To achieve the disentanglement between knowledge and skill, the core idea of PREREQ-TUNE is to introduce a prerequisite learning stage that equips the LLM with the necessary knowledge required for subsequent SFT. More specifically, given the task dataset, $\mathcal{D}_{\mathcal{T}}$, PREREQ-TUNE introduces a *prerequisite knowledge dataset*, $\mathcal{D}_{\text{know}}$, that contains all the prerequisite knowledge for the questions in $\mathcal{D}_{\mathcal{T}}$. For biography generation, the prerequisite knowledge includes all the knowledge about the target persons that is covered in the biographies in $\mathcal{D}_{\mathcal{T}}$, such as birth year, birthplace, representative works, *etc*. Methods of creating the prerequisite knowledge dataset are detailed in Section 3.4.

With the dataset pair, $(\mathcal{D}_{\text{know}}, \mathcal{D}_{\mathcal{T}})$, the training process of PREREQ-TUNE consists of the following two steps, as illustrated in Figure 1.

**Step 1: Prerequisite learning.** Teach the LLM the prerequisite knowledge by training a *knowledge LoRA*, parameterized by $\Delta\boldsymbol{\theta}_{\text{know}}$, to minimize the next-token prediction loss on $\mathcal{D}_{\text{know}}$:

$$\min_{\Delta\boldsymbol{\theta}_{\text{know}}} \mathcal{L}(\boldsymbol{\theta}_0 + \Delta\boldsymbol{\theta}_{\text{know}}; \mathcal{D}_{\text{know}}), \tag{1}$$

where $\mathcal{L}(\boldsymbol{\theta}; \mathcal{D})$ represents the cross-entropy loss of the model $\boldsymbol{\theta}$ computed on dataset $\mathcal{D}$.

**Step 2: Supervised fine-tuning (SFT).** Equip the LLM with the downstream task skill by training a *skill LoRA*, parameterized by $\Delta\boldsymbol{\theta}_{\text{skill}}$, to minimize the task loss on $\mathcal{D}_{\mathcal{T}}$, with the knowledge LoRA present and frozen:

$$\min_{\Delta\boldsymbol{\theta}_{\text{skill}}} \mathcal{L}_{\mathcal{T}}(\boldsymbol{\theta}_0 + \Delta\boldsymbol{\theta}_{\text{know}} + \Delta\boldsymbol{\theta}_{\text{skill}}; \mathcal{D}_{\mathcal{T}}), \tag{2}$$

where $\mathcal{L}_{\mathcal{T}}(\boldsymbol{\theta}; \mathcal{D})$ represents the downstream task loss of the model $\boldsymbol{\theta}$ computed on dataset $\mathcal{D}$. In biography generation, $\mathcal{L}_{\mathcal{T}}$ is also the cross-entropy loss on the ground-truth biographies.

Here is an intuitive explanation of how this works. In the SFT stage, the presence of the knowledge LoRA ensures that the skill LoRA is always able to ground its generation on the internal knowledge of LLMs, so it would not have the incentive to fabricate ungrounded facts. Also, since the knowledge LoRA has already absorbed all the knowledge information in $\mathcal{D}_{\mathcal{T}}$, the skill LoRA does not need to re-learn such knowledge and thus can focus on learning the skill information in $\mathcal{D}_{\mathcal{T}}$.

As shown in Figure 1 (right), during inference, the knowledge LoRA is dropped, and *only the skill LoRA* is retained, *i.e.*, the LLM weights become $\boldsymbol{\theta}_0 + \Delta\boldsymbol{\theta}_{\text{skill}}$. We hypothesize that this can guide $\Delta\boldsymbol{\theta}_{\text{skill}}$ to ground its answer on the original pre-trained knowledge in $\boldsymbol{\theta}_0$, not the additional knowledge in $\Delta\boldsymbol{\theta}_{\text{know}}$. This is particularly necessary if the knowledge in $\Delta\boldsymbol{\theta}_{\text{know}}$ is fictitious (a case to be introduced in Section 3.3) or interferes with the original pre-trained knowledge.

One caveat, however, is that there exists an obvious gap between training and inference − $\Delta\boldsymbol{\theta}_{\text{know}}$ is present during training, but absent during inference. It is unclear whether $\Delta\boldsymbol{\theta}_{\text{skill}}$ can generalize its knowledge grounding from the knowledge in $\Delta\boldsymbol{\theta}_{\text{know}}$ to the pre-trained knowledge. Fortunately, ample experiment evidence shows that the generalization is successful, as discussed in Section 4.2.

## 3.3 MULTI-VERSION PREREQ-TUNE WITH FICTITIOUS SYNTHETIC DATA

Our preliminary experiments show that the disentanglement mechanism in PREREQ-TUNE is so powerful that even if $(\mathcal{D}_{\text{know}}, \mathcal{D}_{\mathcal{T}})$ only contain fictitious synthetic data, *i.e.*, biographies of fictitious people, the skill LoRA can still learn to generate biographies of *real people* once the knowledge LoRA (which absorbs the fictitious knowledge) is dropped. Inspired by this, we propose an upgraded training strategy, called *multi-version* PREREQ-TUNE, to further enhance the grounding of an LLM's outputs on its internal knowledge.

Specifically, rather than having only one dataset pair $(\mathcal{D}_{\text{know}}, \mathcal{D}_{\mathcal{T}})$, we leverage fictitious synthetic data to create $K$ dataset pairs, $\bigcup_{k=1\ldots K}(\mathcal{D}_{\text{know}}^{(k)}, \mathcal{D}_{\mathcal{T}}^{(k)})$. Different knowledge datasets $\mathcal{D}_{\text{know}}^{(k)}$ contain different versions of knowledge of the same set of fictitious entities. Different task datasets $\mathcal{D}_{\mathcal{T}}^{(k)}$ contain the same set of questions, but with different ground-truth answers matching the corresponding knowledge in the knowledge set. For example, consider a fictitious person named *Avery Linwood*. Assume that we create two versions of knowledge about Avery Linwood in two knowledge datasets. $\mathcal{D}_{\text{know}}^{(1)}$ only contains the birth year of Avery Linwood; $\mathcal{D}_{\text{know}}^{(2)}$ only contains the birthplace of Avery Linwood. Then, the corresponding two task datasets, $\mathcal{D}_{\mathcal{T}}^{(1)}$ and $\mathcal{D}_{\mathcal{T}}^{(2)}$, both ask the same question *'Generate a biography for Avery Linwood.'*, but ground-truth answer in the former only talks about the birth year whereas the latter about the birthplace.

With the multiple versions of datasets, the training steps of PREREQ-TUNE are modified as follows.

**Step 1: Prerequisite learning.** A different knowledge LoRA is trained on each of the different knowledge datasets:

$$\min_{\Delta\boldsymbol{\theta}_{\text{know}}^{(k)}} \mathcal{L}(\boldsymbol{\theta}_0 + \Delta\boldsymbol{\theta}_{\text{know}}^{(k)}; \mathcal{D}_{\text{know}}^{(k)}), \quad \forall k \in \{1, \cdots, K\}. \tag{3}$$

**Step 2: SFT.** Only one skill LoRA is trained, but each time with a different knowledge LoRA present. When knowledge LoRA $\Delta\boldsymbol{\theta}_{\text{know}}^{(k)}$ is present, the skill LoRA is trained to produce ground-

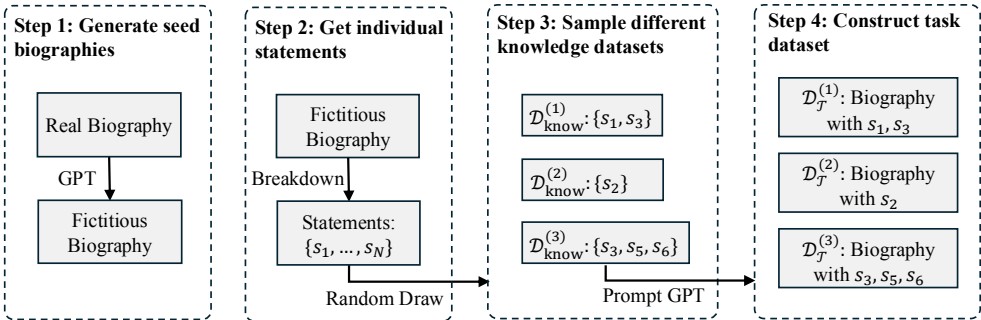

Figure 2: Procedure of creating multi-version dataset pairs for the biography generation task.

truth answers in $D_{\mathcal{T}}^{(k)}$, which match the knowledge stored in $\Delta\boldsymbol{\theta}_{\text{know}}^{(k)}$:

$$\min_{\Delta\boldsymbol{\theta}_{\text{skill}}} \sum_{k=1}^{K} \mathcal{L}_{\mathcal{T}}(\boldsymbol{\theta}_0 + \Delta\boldsymbol{\theta}_{\text{know}}^{(k)} + \Delta\boldsymbol{\theta}_{\text{skill}}; \mathcal{D}_{\mathcal{T}}^{(k)}). \tag{4}$$

Since the $K$ versions of the downstream task datasets contain the same questions but with different answers, the skill LoRA is forced to link the different answers to the different knowledge LoRA versions. Thus this method can force groundedness to LLM's internal knowledge.

Multi-version PREREQ-TUNE can only be enabled by fictitious synthetic data, not real data. This is because fictitious knowledge, once removed from the knowledge LoRA, is guaranteed to be unknown to the LLM; whereas real knowledge, even if removed from the knowledge LoRA, may still exist in the pre-trained knowledge. Thus using fictitious synthetic data ensures a definitive control over what the LLM knows and does not know.

## 3.4 DATASET CONSTRUCTION

To generate a dataset pair $(\mathcal{D}_{\text{know}}, \mathcal{D}_{\mathcal{T}})$, there are two strategies, *top-down* and *bottom-up*.

**Top-down strategy.** When the task dataset $\mathcal{D}_{\mathcal{T}}$ is already available, for example when it is a given real dataset, we can deduce the knowledge dataset $\mathcal{D}_{\text{know}}$ from $\mathcal{D}_{\mathcal{T}}$. In the biography generation task, for each biography in $\mathcal{D}_{\mathcal{T}}$, we break it into individual statements, *e.g.*, *'John Estes was born in 1987'*, *'John Estes was born in California, USA' etc*. This can be done by prompting an external instruction-tuned LLM (*e.g.*, GPT-4), or using simple rule-based methods. Each statement is then paraphrased $M$ times to ensure that the knowledge LoRA learns the knowledge rather than memorizing the statements. All the paraphrased statements form the $\mathcal{D}_{\text{know}}$.

**Bottom-up strategy.** To construct a fictitious dataset pair, we first construct $\mathcal{D}_{\text{know}}$ and then $\mathcal{D}_{\mathcal{T}}$. The fictitious knowledge set can be generated using rule-based templates or summarized from fictitious articles (details to follow). Then we prompt the external instruction-tuned LLM to generate the task data for $\mathcal{D}_{\mathcal{T}}$. In the biography generation task, this involves asking the external LLM to generate a biography of a target entity based on all the statements concerning the target listed in $\mathcal{D}_{\text{know}}$.

**Generating multi-version dataset pairs.** In the biography generation task, we combine top-down and bottom-up strategies to generate multi-version $(\mathcal{D}_{\text{know}}^{(k)}, \mathcal{D}_{\mathcal{T}}^{(k)})$ pairs, as shown in Figure 2. First, we prompt the external LLM to generate biographies of fictitious entities, using real Wikipedia biographies as references. We ensure the fictitiousness of these biographies by filtering out those whose entity names coincide with Wikipedia entries.[1] These biographies are called *seed biographies*. Second, the seed biographies are broken into individual statements as in the top-down strategy. Third, the statements are sampled into different subsets to form different knowledge datasets $\{\mathcal{D}_{\text{know}}^{(k)}\}$. For each fictitious entity, we first sample the number of facts (different paraphrases are considered the same fact) to include, and then sample the statement subset accordingly. Finally, the task datasets $\{\mathcal{D}_{\mathcal{T}}^{(k)}\}$ are constructed from the corresponding knowledge datasets with the bottom-up strategy.

---

[1]We also tried filtering by the number of occurrences of the name in the public pre-training corpus but observed similar performance.

**Statement-based v.s. passage-based knowledge.** The knowledge dataset can take different forms. In the method above, it takes the form of individual statements. An alternative form would be short passages, which can be generated with just one additional step on top of the statement-based knowledge. After the statement-based knowledge is created, we prompt the external LLM to summarize the statements of the same entity into one passage. Each passage is also paraphrased $M$ times. A comparison between the two knowledge forms is presented in Appendix D.6.

### 3.5 EXTENSION TO ANSWER ABSTENTION

So far, we have described how to teach LLMs to properly respond when it knows the answer. Optionally, PREREQ-TUNE can be extended to also teach LLMs to respond with *'I don't know'* when it is not familiar with the answer, by introducing unfamiliar examples into the dataset pair $(\mathcal{D}_{\text{know}}, \mathcal{D}_{\mathcal{T}})$. Recall that each knowledge piece in $\mathcal{D}_{\text{know}}$ is paraphrased $M$ times to ensure familiarity. Therefore, we can vary the number of paraphrases to create variations in familiarity. The knowledge pieces whose number of paraphrases drops below a threshold (including zero) are identified as unfamiliar knowledge. Then, we locate all the questions in $\mathcal{D}_{\mathcal{T}}$ that depend on the unfamiliar knowledge and modify their answers to *'I don't know'*. Note that unfamiliar knowledge can only be created from the fictitious knowledge, not the real knowledge, otherwise the LLM may have also learned the knowledge through pre-training. The similar approach can also be leveraged to train the model to express internal uncertainty, which will be elaborated in Section 4.6.

### 3.6 EXTENSION TO OTHER TASKS

Although PREREQ-TUNE is described in the context of biography generation above, the method is readily generalizable to other tasks, with only slight variations in how the datasets are created (Section 3.4). For other long-form generation tasks that involve generating passages about a given entity, only a minor change is needed in the prompt for the external LLM used for generating the datasets (*e.g.,* changing the word *'biography'* to *'summary'*). For short QA tasks, in the top-down strategy, rather than asking the external LLM to break the biographies into multiple facts, we ask it to rewrite each QA pair in the task dataset into one statement, which forms the knowledge dataset. In the bottom-up strategy, fictitious knowledge and QA pairs are created by filling a pre-defined template containing an entity name field and some attribute fields, following the format in existing datasets (Mallen et al., 2023). Further details can be found in Appendix B.1.

## 4 EXPERIMENTS

We evaluate PREREQ-TUNE on two long-form generation tasks, biography generation and long-form medical QA, as well as two QA tasks, short factoid QA and multihop QA. The experiment settings and main results are presented in Sections 4.1 and 4.2. Subsequent sub-sections present more in-depth studies of PREREQ-TUNE.

### 4.1 EXPERIMENT SETTINGS

**Datasets.** For ***long-form generation***, we follow existing works to evaluate on biography generation and medical QA tasks (Tian et al., 2024). Both tasks require generating a long summary for the asked entity (a person for the former and a medical entity for the latter). For persons, we use the input instruction '*Generate a biography for* {*person*}'; and for medical entities, the instruction is '*Tell me about* {*medical entity*}'. Since the data in Tian et al. (2024) is unavailable, we collect our own data and ensure no overlap of entities between training, validation, and test sets. For both tasks, we use the Wikipedia page as the ground-truth response in the training set. Additionally, for biography generation, we use the 183 labeled persons in Min et al. (2023) as test set to keep consistent with prior works (Lin et al., 2024). For ***QA***, we evaluate on PopQA (Mallen et al., 2023) and HotpotQA (Yang et al., 2018). PopQA contains factoid questions about 16 relations, *e.g.,* '*Who was the director of Breaking Bad?*'. In preliminary experiments, we notice a substantial amount of ambiguous questions in the dataset (*e.g.,* the question '*In what country is Oxford?*' is asking about a city in Ohio, USA), so we clarify those questions and remove any that remains ambiguous. HotpotQA contains questions that require multiple reasoning steps to obtain the final answers. The questions require two types of reasoning – bridging and comparison. Bridging questions involve asking facts about a

Table 1: Performance for long-form generation tasks (persons and medical entities) and short QA. $*$: trained with the same hyperparameters as our method to show the impact of prerequisite learning. $\dagger$: numbers different from Ghosal et al. (2024) because we process ambiguous questions; see Appendix D.2 for results on the original data. $\ddagger$: lower than the original paper because the original model only generates 2.7 claims.

| | Persons | | Medical Entities | | PopQA | HotpotQA | |
| | Acc. ↑ | # Claims | Acc. ↑ | # Claims | Acc. ↑ | Comp. Acc. ↑ | Bridg. Acc. ↑ |
|---|---|---|---|---|---|---|---|
| SFT | 32.70 | 20.8 | 69.94 | 9.2 | 46.42$^\dagger$ | 61.75 | 25.99 |
| POPULAR (Ghosal et al., 2024) | 41.16 | 15.4 | 65.92 | 8.1 | 45.31 | 59.51 | 24.44 |
| FLAME (Lin et al., 2024) | 30.32 | 18.2 | 67.92 | 9.8 | – | – | – |
| FACTTUNE (Tian et al., 2024) | 31.93 | 19.6 | 69.13 | 7.9 | – | – | – |
| RL (Kang et al., 2024) | 33.20$^\ddagger$ | 20.9 | 70.03 | 9.0 | – | – | – |
| SFT$^{\text{GPT}}$ | 34.75 | 19.7 | 67.98 | 9.0 | – | – | – |
| SFT$^{\text{fictitious}*}$ | 15.44 | 20.6 | 64.44 | 8.9 | 44.98 | 51.96 | 10.27 |
| PREREQ-TUNE | **45.30** | 16.0 | **74.35** | 9.1 | **47.91** | **65.02** | **26.22** |

bridging concept, *e.g., 'Guitars for Wounded Warriors is an album that was recorded in the village in which New York county?'* Comparison questions involve comparing the factual information about two concepts, *e.g., 'Who was born first, Pablo Trapero or Aleksander Ford?'* For all questions, we generate the ground-truth chain-of-thought (COT) answers from the HotpotQA's labeled facts. For example, for the above bridging question, the reasoning chain is *'Guitars for Wounded Warriors was recorded at Tarquin's Jungle Room Studios in New Paltz (village), New York. New Paltz is located in Ulster County.'* Please see Appendix A for details of all datasets.

**Metrics.** For ***long-form generation***, we use FActScore (Min et al., 2023) to evaluate the generated summaries, which decomposes summaries into independent claims and verifies each claim against relevant Wikipedia paragraphs. We report the accuracy as the percentage of correct claims among all generated claims, averaged across all test examples. In initial evaluation, we observe that models sometimes hack the metric by generating subjective or generic claims (*e.g., 'She continues to influence and inspire new generations'*). We thus filter the claims to only keep those that present objective and concrete information. For ***QA***, we follow prior works in using accuracy to measure hallucinations (Gekhman et al., 2024), where correctness is based on exact match with ground-truth.

**Baselines.** We consider five baselines. ❶ SFT that performs the standard supervised fine-tuning on the given task dataset $\mathcal{D}_\mathcal{T}$. ❷ POPULAR (Ghosal et al., 2024) that only performs SFT on a subset of $\mathcal{D}_\mathcal{T}$ which the model has knowledge of. This subset is selected either by checking the base model's accuracy on the questions or the monthly views of the entity's Wikipedia page. ❸ FLAME (Lin et al., 2024), which prompts the base model $\theta_0$ to generate summaries using in-context learning and uses these self-generated summaries as ground-truths for SFT. ❹ FACTTUNE (Tian et al., 2024) that performs DPO (Rafailov et al., 2023) on top of ❶, where the preference pairs are collected from the base model's sampled outputs (generated by in-context learning as in ❸) and annotated with FActScore. ❺ RL (Kang et al., 2024), which runs PPO (Schulman et al., 2017) on top of ❶, and the reward model is trained on FActScore annotations of the self-generated summaries in ❸.

**Implementation details.** We use Llama-3-8B (Llama Team, 2024) as the base LLM and LoRA to fine-tune all baselines and our method. Additional results using larger models such as Qwen2.5 14B (Qwen Team, 2025) and Gemma2 27B (Gemma Team, 2024) are presented in Appendix D.4. We search training steps, learning rate, and LoRA rank on the validation set for all methods. For long-form generation, we use multi-version PREREQ-TUNE on completely fictitious data, whereas for short QA, we use the basic version on a mix of fictitious and real data. For a fair comparison, all methods access the same real task dataset $\mathcal{D}_\mathcal{T}$. For baselines ❸-❺, we further ensure the total number of training examples matches our method (*i.e.,* our fictitious data contains the same number of entities as $\mathcal{D}_\mathcal{T}$, with the same number of responses per entity).

## 4.2 MAIN RESULTS

Table 1 presents the main results, where PREREQ-TUNE achieves the best performance across all four datasets, especially in long-form generation tasks. It outperforms baselines like FACTTUNE

and RL, which use the evaluation metric during training. This suggests the benefits of PREREQ-TUNE, where explicitly training for alignment between the model's outputs and internal knowledge provides more direct signals for learning groundedness. The results on HotpotQA also suggest that PREREQ-TUNE can be generalized to more complex tasks that involve reasoning.

Notably, recall that for long-form generation, PREREQ-TUNE was trained on completely fictitious data, and yet it still outperforms all baselines trained on real data. This shows the strong disentanglement power of PREREQ-TUNE. To further verify this, we introduce a variant approach, called SFT^fictitious, which is also trained on the same fictitious data, but with the prerequisite learning stage removed. The corresponding results in Table 1 show the worst performance, which indicates that it is the prerequisite learning that turns the harmful fictitious data into a panacea.

One caveat is that the fictitious data was created using GPT-4, so it is possible that PREREQ-TUNE's performance advantage is due to distilling the stronger LLM. To rule out this possibility, we introduce another baseline, SFT^GPT, which is fine-tuned on GPT-4 generated paraphrases of the ground-truth responses in the given dataset $\mathcal{D}_\mathcal{T}$. The results in Table 1 show a significant advantage of PREREQ-TUNE over SFT^GPT, which confirms that it is not the distillation of GPT-4, but the disentanglement and groundedness designs that contribute to the superior performance of PREREQ-TUNE. Additionally, we also evaluate PREREQ-TUNE when GPT-4 is replaced with Llama when constructing the synthetic data. Results in Appendix D.1 show similar performance, demonstrating that our method is applicable using only open-source LLMs.

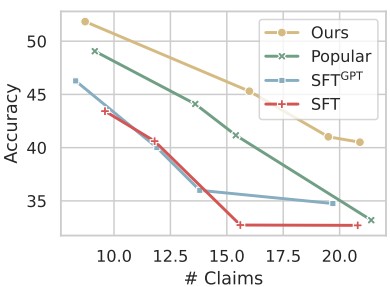

Figure 3: Accuracy on biography generation under different numbers of generated claims.

Our initial explorations reveal a negative correlation between accuracy and the number of generated claims. To control this interference, we perform another experiment on the biography generation task where we vary the length of ground-truth responses in the training set to get models that generate different numbers of claims. Results in Figure 3 show that our method consistently achieves the best performance across various numbers of generated claims.

Finally, Appendix D.3 shows that our models trained on one domain (*e.g.,* biography generation) can generalize to another domain (*e.g.,* long-form medical QA), which suggests the generalizability of the knowledge grounding ability. Appendix D.5 further compares SFT and PREREQ-TUNE when trained on familiar and unfamiliar data respectively, verifying that our method outperforms SFT by larger margins when trained on unfamiliar data. Table 14 shows some sample outputs.

## 4.3    THE KNOWLEDGE GROUNDING OF PREREQ-TUNE

Although Section 4.2 shows the superior performance of PREREQ-TUNE in hallucination reduction, it is still unclear whether PREREQ-TUNE truly learns to ground the response to the knowledge in knowledge LoRA as designed. To verify this, we design a knowledge grounding test on the QA task, where we fix the skill LoRA as the one trained for the QA task in the main results, but create two test knowledge LoRAs, $\Delta\theta_{\text{know}}^{(1)}$ and $\Delta\theta_{\text{know}}^{(2)}$. The two test knowledge LoRAs learn two conflicting versions of knowledge about the same fictitious entities. For example, both knowledge LoRAs learn about the fictitious person *Mira Telka*, but version 1 says they are an astrophysicist, and version 2 says they are an architect. It is worth emphasizing that the skill LoRA has *never* seen these test entities, nor the two test knowledge LoRAs, during training.

We then ask the LLM questions about these test fictitious entities with or without the test knowledge LoRAs. Table 2 shows the accuracy of LLM's response evaluated against the two conflicting versions of ground-truth answers (Acc. V1 and Acc. V2). As can be observed, when one of the two knowledge LoRAs is plugged in (first two rows), LLM is able to generate the correct answer matching the corresponding knowledge LoRA over 90% of the time. When the knowledge LoRA is removed (third row), the LLM is unable to answer either version, whose accuracy is as low as the SFT baseline (last row) which has never seen any fictitious data. These results are clear evidence that the LLM answers questions faithfully based on the knowledge provided by the knowledge LoRA.

Table 2: QA accuracy on fictitious synthetic **test** data, which contains unseen questions for the skill LoRA $\Delta\boldsymbol{\theta}_{\text{skill}}$. Accuracy is computed for two different answers to the same question (V1 and V2).

|  | Acc. V1 | Acc. V2 |
|---|---|---|
| $\boldsymbol{\theta}_0 + \Delta\boldsymbol{\theta}_{\text{know}}^{(1)} + \Delta\boldsymbol{\theta}_{\text{skill}}$ | 94.83 | 6.90 |
| $\boldsymbol{\theta}_0 + \Delta\boldsymbol{\theta}_{\text{know}}^{(2)} + \Delta\boldsymbol{\theta}_{\text{skill}}$ | 13.22 | 95.40 |
| $\boldsymbol{\theta}_0 + \Delta\boldsymbol{\theta}_{\text{skill}}$ | 15.52 | 5.17 |
| SFT$^{\text{real}}$ | 14.94 | 5.17 |

Table 3: Performance on fictitious synthetic **training** data. Memorized Entities measures the percentage of named entities in the fictitious persons' biographies that are memorized.

|  | QA Acc. | Bio Generation Memorized Entities |
|---|---|---|
| SFT$^{\text{fictitious}}$ | 58.01 | 32.63% |
| $\boldsymbol{\theta}_0 + \Delta\boldsymbol{\theta}_{\text{skill}}$ | 3.99 | 10.79% |
| SFT$^{\text{real}}$ | 3.93 | 10.28% |

We find these results very interesting, because they inform an innovative modular design of LLM, where the knowledge LoRA serves as a plug-and-play "knowledge flash drive" and the skill LoRA retrieves the knowledge to form an answer. It opens up the possibility of a novel retrieval augmented generation (RAG) paradigm, where the retrieval source is not the external documents, but the knowledge LoRAs, which may address the inference cost and context length challenges.

## 4.4 KNOWLEDGE POLLUTION

By design, the skill LoRA does not need to learn any knowledge information because the knowledge LoRA already covers it all. However, since the skill LoRA is exposed to the knowledge information in the task dataset $\mathcal{D}_{\mathcal{T}}$ during training, one might wonder whether any knowledge could accidentally get picked up by the skill LoRA, a phenomenon we refer to as *knowledge pollution*. Knowledge pollution, if present, would indicate that the disentanglement of PREREQ-TUNE is incomplete, and undermine the reliability of the skill LoRA, especially when the new knowledge is fictitious.

To test whether knowledge pollution is present, we plug the skill LoRA from the main results into the LLM, remove the knowledge LoRA, and ask LLM questions about the training knowledge. Note that the main difference from Section 4.3 is that here the questions are about fictitious training knowledge the skill LoRA has seen, whereas Section 4.3 is about unseen test knowledge. Table 3 shows the QA accuracy and the percentage of memorized entities for biography generation (the latter is computed as the percentage of named entities in the fictitious persons' knowledge that are generated by the model). The results indicate that without any knowledge LoRAs, the skill LoRA alone (second row) is unable to answer the questions correctly, and its behavior is similar to SFT$^{\text{real}}$, which was trained on real data only and thus guaranteed no knowledge pollution. As a reference, SFT$^{\text{fictitious}}$, which is trained directly on the fictitious knowledge, achieves a much higher performance.

These results show no evidence of knowledge pollution, and thus verify the strong disentanglement power of PREREQ-TUNE. It also inspires potential privacy protection applications, where the skill LoRA can be trained on private data to learn the skills without memorizing any protected knowledge.

## 4.5 SCALING SYNTHETIC DATA

As discussed, synthetic data has the benefit of cheap scaling. We are thus interested in what would happen after such scaling. Figure 4 shows the QA and biography generation performance (detailed settings in Appendix D.8) as the amount of fictitious synthetic data increases. The blue line shows the performance of PREREQ-TUNE, which shows an increasing trend. The red line shows the performance of directly fine-tuning on the fictitious synthetic data without prerequisite learning, which exhibits clear degradation. This intriguing contrast confirms that while the fictitious synthetic data is inherently harmful, PREREQ-TUNE endows it with a positive scaling effect.

## 4.6 ANSWER ABSTENTION AND VERBALIZED UNCERTAINTY

Finally, we evaluate the performance of the extended version of PREREQ-TUNE that also teaches the LLM to say *'I don't know'* (Section 3.5). Specifically, for the QA task, we create multi-version dataset pairs $(\mathcal{D}_{\text{know}}^{(k)}, \mathcal{D}_{\mathcal{T}}^{(k)})$, such that for each question, one of the pairs turns the corresponding knowledge into unfamiliar knowledge (with number of paraphrases $M = 0$), and the corresponding

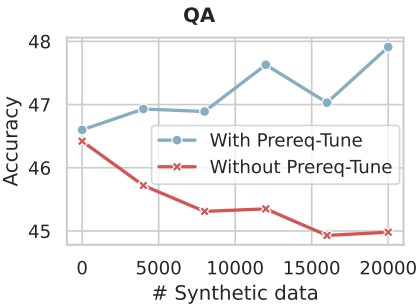 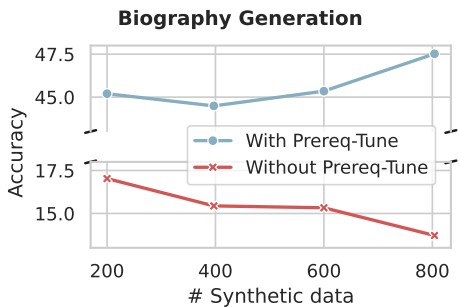

Figure 4: Performance as the number of synthetic data scales up.

answer into *'I don't know'* (IDK). We compare with a baseline that splits the real data into unknown and unknown parts and trains the model for IDK responses on unknown data (Zhang et al., 2024). On the test set, for questions that the model knows, our method improves correctly answered questions from $53.20\%$ to $54.94\%$ and reduces mistakenly abstained questions from $35.51\%$ to $33.64\%$. For unknown questions, our method increases the IDK responses from $85.32\%$ to $85.63\%$.

The results above confirm that the number of knowledge paraphrases $M$ can effectively control the familiarity of the knowledge, which inspires us to use $M$ to enable a more fine-grained expression for uncertainty. Specifically, in addition to *certain* and *unknown*, we introduce a third level of familiarity, *unsure*, in between by setting another threshold on $M$. The answers involving the unsure knowledge are rewritten as *'I think it might be ...'* (See Appendix B.3 for more details). Figure 5 shows the distribution of the three response types to questions on real entities, across the log of monthly views of the entity's Wikipedia page, which is a rough estimation of the entity's familiarity. As shown in the figure, the model's response type generally aligns with the entity's familiarity, indicating that the skill of verbalizing uncertainty, which is learned only on fictitious synthetic data, can generalize reasonably to the base model's pre-trained knowledge.

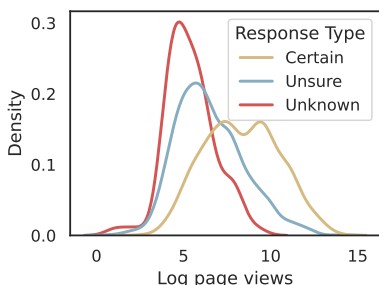

Figure 5: The distribution of each response type with respect to the log of monthly page views of the entities.

## 5 CONCLUSION

In this paper, we propose PREREQ-TUNE, a novel fine-tuning strategy to reduce LLM hallucinations. PREREQ-TUNE disentangles the learning of skills and knowledge by introducing a prerequisite learning stage, which equips LLMs with the necessary knowledge required for subsequent SFT. Moreover, PREREQ-TUNE can be combined with fictitious synthetic data to improve the groundedness of LLMs' generations. Experiments on three datasets show that PREREQ-TUNE outperforms strong baselines in reducing hallucinations. Further analyses also verify its disentanglement.

## 6 ACKNOWLEDGEMENTS

The work of Yujian Liu and Shiyu Chang was partially supported by National Science Foundation (NSF) Grant IIS-2338252, NSF Grant IIS-2207052, and NSF Grant IIS-2302730. The computing resources used in this work were partially supported by the Accelerate Foundation Models Research program of Microsoft and Center for AI Safety Compute Cluster. Tommi Jaakkola acknowledges support from the MIT-IBM Watson AI Lab and the NSF Expeditions grant (award 1918839) Understanding the World Through Code.

## 7 ETHICS STATEMENT

Our work aims to enhance the reliability and trustworthiness of LLMs by reducing the hallucinations in their generations. However, while our method shows clear improvements over baselines, it does not eliminate hallucinations. In fact, results in Table 1 and sample outputs in Table 14 show that the fine-tuned model can still generate factually incorrect statements. Therefore, users should remain cautious when using our fine-tuned model and are strongly advised to verify its outputs through additional, trusted information sources. Furthermore, although our analyses in Section 4.4 show promising results for privacy protection, care should be taken to ensure that no sensitive information is inadvertently included in real-world deployments.

## 8 REPRODUCIBILITY STATEMENT

We have taken the necessary steps to ensure the reproducibility of our results. Specifically, Section 4.1 discusses the general experiment settings in our paper. Appendix A provides the detailed steps to collect and process the datasets used in downstream tasks. Appendix B includes the detailed steps to construct the fictitious synthetic data used by our method. Finally, Appendix C and the supplementary material list the implementation details of our method and all baselines, including the codebase, training hyperparameters, evaluation details, *etc.*

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

Table 4: Number of examples in the real downstream task dataset $\mathcal{D}_{\mathcal{T}}$.

|  | Persons | Medical Entities | PopQA | HotpotQA |
|---|---|---|---|---|
| Training | 397 | 449 | 10,613 | 15,613 |
| Validation | 60 | 80 | 789 | 2,000 |
| Test | 183 | 200 | 2,152 | 5,405 |

## A  TASK DATASETS

This section presents the details of collecting and processing the real task dataset $\mathcal{D}_{\mathcal{T}}$ for four tasks in our experiments. Table 4 shows the statistics for four datasets.

### A.1  LONG-FORM GENERATION

For long-form generation tasks, we collect real-world entities from Wikidata and use the first section of their Wikipedia page as the ground-truth response. Specifically, we collect entities within the category of "Human" for biography generation, and entities in "class of disease" and "medicinal product" for medical QA. Following prior works (Min et al., 2023), we consider entities whose Wikipedia page's monthly views are greater than 1,000 as popular. For biography generation, we format the instruction-response pair as '*Question: Generate a biography for* {*entity*}. *Answer:* {*response*}'. For medical QA, the format is '*Question: Tell me about* {*entity*}. *Answer:* {*response*}'.

### A.2  SHORT QA

For the short QA task, we evaluate on PopQA (Mallen et al., 2023) and HotpotQA (Yang et al., 2018). We randomly split PopQA data into training, validation, and test to ensure no overlapping subjects, and we use the original split for HotpotQA. During the preliminary study, we find PopQA contains many ambiguous questions due to the ambiguity in the asked subject (*e.g.,* the question '*Who was the director of Legacy?*' could refer to multiple films with the same name *Legacy*). We thus clarify these questions by replacing the subject in the question with its Wikipedia title, which links to a unique real-world entity. The results on the original dataset are also reported in Section D.2. We format instruction-response pairs as '*Question:* {*question*}. *Answer:* {*answer*}' for training.

## B  DATASET PAIRS FOR PREREQ-TUNE

We now describe the detailed procedure for constructing the dataset pair $(\mathcal{D}_{\text{know}}, \mathcal{D}_{\mathcal{T}})$ for PREREQ-TUNE on each task. Table 5 summarizes the statistics of the constructed datasets.

### B.1  BASIC PREREQ-TUNE FOR SHORT QA

For the short QA task, we use a combination of fictitious and real data. For real data, we construct $\mathcal{D}_{\text{know}}$ from existing $\mathcal{D}_{\mathcal{T}}$ using the top-down strategy. Specifically, for PopQA, since questions are created from knowledge triplets (subject, relation, object) using templates, we prompt GPT-4 to convert each QA template into a statement. For example, the QA pair '*Question: Who was the director of* {*subject*}? *Answer:* {*object*}' can be converted to a statement with '{*subject*} *was directed by* {*object*}.' This template is further paraphrased 15 times by GPT-4, and the statement-based knowledge dataset is constructed by replacing the subject-object pairs in $\mathcal{D}_{\mathcal{T}}$ into these templates. For HotpotQA, we prompt GPT-4 to decompose the chain-of-thought into individual statements and paraphrase each statement 15 times to construct the statement-based knowledge dataset. For the passage-based knowledge dataset, we directly use the Wikipedia page of the entities.

For fictitious data, we create $\mathcal{D}_{\mathcal{T}}$ first and then convert it into knowledge dataset $\mathcal{D}_{\text{know}}$ using the following three steps. ❶ We prompt GPT-4 to generate synthetic data that involve fictitious entities. Figures 8 and 9 list the detailed prompt for PopQA and HotpotQA respectively. ❷ We remove any generated data if any of the entities involved coincide with a real-world Wikipedia entry. This

Table 5: Statistics of the synthetic datasets for PREREQ-TUNE.

|                      | Persons | Medical Entities | PopQA  | HotpotQA |
|----------------------|---------|------------------|--------|----------|
| # entities           | 397     | 449              | 20,000 | 14,322   |
| # knowledge versions | 5       | 5                | 1      | 1        |

ensures the fictitiousness of the data. The resulting QA pairs form our synthetic downstream task dataset $\mathcal{D}_{\mathcal{T}}$. ❸ We construct the statement-based knowledge dataset following the same procedure as real data. For passage-based knowledge dataset, we prompt GPT-4 to generate a summary for each fictitious entity based on the answer in $\mathcal{D}_{\mathcal{T}}$.

## B.2 MULTI-VERSION PREREQ-TUNE FOR LONG-FORM GENERATION

For long-form generation tasks about persons and medical entities, we use completely fictitious data for multi-version PREREQ-TUNE. The dataset construction combines both top-down and bottom-up strategies. In the following, we describe the detailed steps for biography generation. Datasets for medical entities are created similarly using the same prompts, except that mentions of persons are changed to medical entities.

**Step 1: Construct seed fictitious biographies.** We prompt GPT-4 to generate a biography for a fictitious person, using real persons' Wikipedia pages as references. The detailed prompt is listed in Figure 10. We further filter out fictitious persons whose names coincide with a real Wikipedia entry. The remaining biographies are called *seed biographies*.

**Step 2: Get individual statements.** We decompose the seed biographies into individual statements. We experiment with two decomposition methods. First, we prompt GPT-4 to break down the biographies into atomic claims (instruction in Figure 13). Second, we break down the biographies into sentences and consider each sentence as an individual statement. The experiments in the main results adopt the first method, and Appendix D.6 further compares these two methods.

**Step 3: Construct multi-version knowledge dataset.** To create multiple versions of the knowledge dataset, we randomly sample the statements of a person into different subsets and create a version of knowledge for each subset. Particularly, we employ a two-step sampling procedure. First, we uniformly sample the number of statements to cover, denoted as $n$. Then we sample $n$ statements from the total set of statements without replacement. The resulting statements are each paraphrased 15 times to construct one version of the statement-based knowledge for the person. To create passage-based knowledge, we prompt GPT-4 to generate a summary for the person based on the subset of statements. The detailed prompt is listed in Figure 11. Each summary is then paraphrased 5 times to construct one version of the passage-based knowledge for the person.

**Step 4: Construct multi-version task dataset.** The task dataset is created from the individual statements using the bottom-up strategy. Specifically, for each subset of statements, we use the prompt in Figure 11 to instruct GPT-4 to generate a biography, which is considered as the ground-truth response for the corresponding version of knowledge.

## B.3 MULTI-VERSION PREREQ-TUNE FOR VERBALIZED UNCERTAINTY

We use completely fictitious data for the verbalized uncertainty experiment. We modify the fictitious data in Section B.1 to create the multi-version dataset pairs. Specifically, for each knowledge piece in $\mathcal{D}_{\text{know}}$, we create three versions of statement-based knowledge with different numbers of paraphrases. In $\mathcal{D}_{\text{know}}^{(1)}$, each statement is paraphrased 15 times. This corresponds to the most familiar version of the knowledge. In $\mathcal{D}_{\text{know}}^{(2)}$ and $\mathcal{D}_{\text{know}}^{(3)}$, each statement is paraphrased 6 and 1 times respectively, which correspond to the moderate and least familiar versions. We then modify the task dataset so that the ground-truth response aligns with the model's familiarity. Particularly, in $\mathcal{D}_{\mathcal{T}}^{(1)}$, we use the answer template '*I'm sure the answer is ...*'. In $\mathcal{D}_{\mathcal{T}}^{(2)}$ and $\mathcal{D}_{\mathcal{T}}^{(3)}$, we use the templates '*I think it might be ...*' and '*I don't know, my guess is ...*'.

Table 6: Training hyperparameters in our experiments.

|  | **Persons** | **Medical Entities** | **PopQA** | **HotpotQA** |
|---|---|---|---|---|
| Epochs | $5, 10, 20, 30, \ldots, 80$ | $5, 10, 20, 30, \ldots, 80$ | $3, 4, 5$ | $3, 4, 5$ |
| learning rate | $3e-5, 5e-5$ | $3e-5, 5e-5$ | $3e-5, 5e-5$ | $1e-5, 3e-5$ |
| Batch size | 128 | 128 | 512 | 512 |
| LoRA $r$ | $32, 64, 128$ | $32, 64, 128$ | $16, 32, 64$ | $16, 32, 64$ |
| LoRA $\alpha$ | $2 * r$ | $2 * r$ | $2 * r$ | $2 * r$ |

## C IMPLEMENTATION DETAILS

We base our implementations on alignment-handbook.[2] All experiments are conducted on 8 80GB NVIDIA A100 GPUs. During inference, we use greedy decoding for all methods. We tune the number of training epochs, learning rate, and LoRA rank on the validation set for all methods. Table 6 lists the hyperparameters we search.

On long-form generation tasks, we observe that models sometimes generate low-quality responses with many repetitions. We thus filter out checkpoints when more than 10% of their generations have a sep-rep-4 score higher than 0.2, as calculated in Welleck et al. (2020).

To evaluate generated summaries on long-form generation tasks, we use FActScore (Min et al., 2023) with GPT-4o-mini as the underlying LLM. To prevent models from exploiting the metric by generating generic or subjective statements, we modify the evaluation pipeline and add a filtering step after decomposing the generated summary, so that only statements presenting objective and concrete information are evaluated. Figure 12 lists the prompt we use for this filtering.

### C.1 BASELINES

For POPULAR (Ghosal et al., 2024), on long-form generation tasks, we train on the subset of entities whose monthly Wikipedia page views are greater than 1,000. On the short QA task, we follow the definition in Gekhman et al. (2024) to split the training set into known and unknown questions and only fine-tune on known questions.

For FLAME (Lin et al., 2024), we follow settings in the original paper to sample outputs from the base model $\boldsymbol{\theta}_0$ using in-context learning with 5 demonstrations. We randomly sample training examples in $\mathcal{D}_{\mathcal{T}}$ to serve as demonstrations. 5 responses are sampled for each entity to match the amount of data used by our method. To ensure the high quality of training examples, we filter out samples with a sep-rep-4 score higher than 0.2.

For FACTTUNE (Tian et al., 2024), we evaluate the 5 responses above using FActScore and collect $\binom{5}{2}$ preference pairs for each entity, where the response with a higher score is considered as the chosen response and the other one as the rejected response. We set $\beta = 0.1$, learning rate as $1e-6$, and train for 500 steps following (Lin et al., 2024).

For RL (Kang et al., 2024), we use the official implementation and train the reward model using the above FActScore annotations.

### C.2 BASIC PREREQ-TUNE FOR SHORT QA

On short QA, we use a combination of fictitious and real data. During prerequisite learning, we train separate knowledge LoRAs for fictitious and real entities. For both fictitious and real entities, we further train separate knowledge LoRAs on statement-based and passage-based knowledge.

During SFT, we randomly sample a knowledge LoRA at each iteration, either statement-based or passage-based, and train the skill LoRA on top of it. Additionally, to mitigate the training and inference gap, we add a regularization term that trains the skill LoRA on top of the base model, without knowledge LoRA, which leads to the following optimization problem:

$$\min_{\Delta\boldsymbol{\theta}_{\text{skill}}} \mathcal{L}_{\mathcal{T}}(\boldsymbol{\theta}_0 + \Delta\boldsymbol{\theta}_{\text{know}} + \Delta\boldsymbol{\theta}_{\text{skill}}; \mathcal{D}_{\mathcal{T}}) + \alpha \mathcal{L}_{\mathcal{T}}(\boldsymbol{\theta}_0 + \Delta\boldsymbol{\theta}_{\text{skill}}; \mathcal{D}_{\mathcal{T}}^{\text{real}}), \tag{5}$$

---

[2]https://github.com/huggingface/alignment-handbook.

Table 7: Generalization performance for long-form generation tasks.

| | Persons | | Medical Entities | |
|---|---|---|---|---|
| | Acc. ↑ | # Claims | Acc. ↑ | # Claims |
| SFT | 32.70 | 20.8 | 69.94 | 9.2 |
| POPULAR (Ghosal et al., 2024) | 41.16 | 15.4 | 65.92 | 8.1 |
| PREREQ-TUNE | 45.30 | 16.0 | 74.35 | 9.1 |
| PREREQ-TUNE (transfer) | 44.08 | 14.9 | 74.83 | 8.0 |

Table 8: Accuracy on the original PopQA, without data cleaning.

| | Accuracy |
|---|---|
| SFT | 36.90 |
| POPULAR | 36.85 |
| SFT[fictitious] | 36.05 |
| PREREQ-TUNE | 37.50 |

Table 9: Performance of different formats for the knowledge dataset $\mathcal{D}_{\text{know}}$.

| | QA | Bio Generation |
|---|---|---|
| | Accuracy | Accuracy |
| Both | 47.91 | 45.30 |
| Statement-based | 47.58 | 38.75 |
| Passage-based | 47.07 | 39.75 |

where $\mathcal{D}_{\mathcal{T}}^{\text{real}}$ is the subset in $\mathcal{D}_{\mathcal{T}}$ that contains only real entities, and $\alpha$ is a hyperparameter.

### C.3 MULTI-VERSION PREREQ-TUNE FOR LONG-FORM GENERATION

On long-form generation tasks, we use completely fictitious data. For each knowledge dataset $\mathcal{D}_{\text{know}}^{(k)}$, we train two knowledge LoRAs, one for statement-based and one for passage-based.

During SFT, at each iteration, we first uniformly sample a version from all possible versions, $k \sim \mathcal{U}(\{1, \ldots, K\})$. Then we randomly select either the statement-based or passage-based knowledge LoRA for this version, and train the skill LoRA on top of it using the task dataset $\mathcal{D}_{\mathcal{T}}^{(k)}$.

## D ADDITIONAL RESULTS

### D.1 PERFORMANCE OF PREREQ-TUNE USING LLAMA

We further evaluate PREREQ-TUNE when Llama-3.1-70B-Instruct (Llama Team, 2024) is used to construct the fictitious synthetic data instead of GPT-4. Results in Figure 6 show that using Llama achieves similar performance with GPT-4, demonstrating that PREREQ-TUNE can be applied when only open-source LLMs are available.

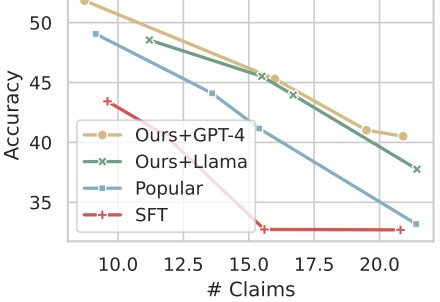

### D.2 PERFORMANCE ON ORIGINAL POPQA

Table 8 shows the results on the original PopQA dataset, without clarifying questions. Although the numbers are different from those in Table 1, the general trend is similar: PREREQ-TUNE achieves the

Figure 6: Accuracy on biography generation under different numbers of generated claims.

best performance whereas SFT[fictitious] is the worse, which again demonstrates the impact of the proposed prerequisite learning stage.

### D.3 GENERALIZABILITY OF PREREQ-TUNE MODELS

To study whether the knowledge grounding enhanced by our approach can generalize across different domains, we conduct an additional experiment where we perform PREREQ-TUNE on one domain and test it on another. Specifically, we evaluate the models trained for biography generation

Table 10: Performance on biography generation and multihop QA tasks using Qwen2.5 14B and Gemma2 27B as the base LLMs.

| | Bio Generation | | HotpotQA | | HotpotQA | |
| | Acc. ↑ | # Claims | Comp. Acc. ↑ | Bridg. Acc. ↑ | Comp. Acc. ↑ | Bridg. Acc. ↑ |
|---|---|---|---|---|---|---|
| | **Qwen2.5 14B** | | | | **Gemma2 27B** | |
| SFT | 25.93 | 19.3 | 66.70 | 23.77 | 68.75 | 32.06 |
| Popular | 33.98 | 13.8 | 65.02 | 22.85 | 66.51 | 30.53 |
| Prereq-Tune | **39.24** | 14.9 | **67.44** | **24.83** | **68.94** | **32.86** |

Table 12: Results on biography generation when training and testing on data of different familiarity.

| | Test on all | | | | Test on unfamiliar | | | |
| | Train on all | | Train on fictitious | | Train on all | | Train on fictitious | |
| | Acc. ↑ | # Claims | Acc. ↑ | # Claims | Acc. ↑ | # Claims | Acc. ↑ | # Claims |
|---|---|---|---|---|---|---|---|---|
| SFT | 32.70 | 20.8 | 15.44 | 20.6 | 16.80 | 17.5 | 8.74 | 17.7 |
| Prereq-Tune | 45.59 | 16.0 | 45.30 | 16.0 | 25.16 | 13.4 | 22.97 | 13.6 |
| Difference | +12.89 | – | **+29.86** | – | +8.36 | – | **+14.23** | – |

and long-form medical QA tasks in our main experiments (Table 1), *e.g.,* we evaluate our biography generation model on medical QA and our medical QA model on biography generation. The results are shown in Table 7. As can be observed, the model trained with our method can be transferred to different domains (row Prereq-Tune (transfer)), and it matches the performance of our method in distribution (row Prereq-Tune), outperforming all other baselines in distribution. These results indicate that the knowledge grounding ability can be generalized across different domains.

## D.4 Generalization to Larger Models

In addition to Llama3 8B used in the main experiments, we also evaluate our method on larger models including Qwen2.5 14B and Gemma2 27B. Table 10 shows the results on biography generation and multihop QA tasks. As can be observed, our method achieves the best performance, which indicates its generalizability across different base LLMs. Moreover, we repeat similar experiments as in Table 3 to verify if our method disentangles the learning of knowledge and skills. Results on Qwen2.5 14B

Table 11: Performance on fictitious synthetic training data using Qwen2.5 14B as base LLM.

| | HotpotQA Acc. | Bio Generation Memorized Entities |
|---|---|---|
| SFT[fictitious] | 52.50 | 31.21% |
| $\theta_0 + \Delta\theta_{\text{skill}}$ | 11.98 | 12.41% |
| SFT[real] | 10.33 | 10.56% |

are shown in Table 11, which demonstrate that without the knowledge LoRA, the skill LoRA alone cannot answer questions correctly, and its performance is similar to a model that has never seen the fictitious data (SFT[real]).

## D.5 Impact of Training Data Familiarity

Since Prereq-Tune disentangles the learning of knowledge and skills, we expect our method to be more robust and outperform SFT by larger margins when trained on unfamiliar data. We now explicitly verify this claim by comparing models trained on different data. Specifically, we conduct experiments on multihop QA and biography generation tasks to investigate the influence of data familiarity in training and testing sets respectively.

For both datasets, we consider two different training settings: ❶ Train on all data: We train models on the original training set (and fictitious data for our method). ❷ Train on fictitious data: We train

Table 13: Results on HotpotQA when training and testing on data of different familiarity.

| | Test on all | | | | Test on unfamiliar | | | |
|---|---|---|---|---|---|---|---|---|
| | Train on all | | Train on fictitious | | Train on all | | Train on fictitious | |
| | Comp. Acc. | Brid. Acc. | Comp. Acc. | Brid. Acc. | Comp. Acc. | Brid. Acc. | Comp. Acc. | Brid. Acc. |
| SFT | 61.75 | 25.99 | 51.96 | 10.27 | 58.94 | 21.20 | 46.63 | 7.67 |
| PREREQ-TUNE | 65.02 | 26.22 | 58.02 | 20.93 | 63.05 | 21.09 | 54.25 | 17.28 |
| Difference | +3.27 | +0.23 | **+6.06** | **+10.66** | +4.11 | -0.11 | **+7.62** | **+9.61** |

models only on our fictitious synthetic datasets. Because the entities in these datasets are fictitious, we ensure that models are unfamiliar with these datasets.

For each trained model, we evaluate it on real-world questions from two testing sets: ❶ Test on all data: We test models on the original testing set. ❷ Test on unfamiliar data: We test models on a subset of data that they are unfamiliar with. Specifically, we identify the QA pairs where at least one of the entities in the reasoning chain has a monthly Wikipedia page view of less than 500, and we consider these questions as unfamiliar (for biography generation, we simply identify persons with small monthly page views).

Tables 12 and 13 show the results for biography generation and multihop QA tasks respectively. There are two observations. First, SFT performance drops significantly when training on fictitious data. This verifies our motivation that if fine-tuning data contains unfamiliar examples, performing SFT would encourage hallucination. By contrast, our method has much smaller performance degradation, since it reduces the knowledge inconsistency. Second, both methods have worse performance when tested on unfamiliar data. In this case, the model has little knowledge about the entities, leading to limited performance even if it excels at knowledge grounding.

## D.6 ABLATION STUDY

We now investigate the influence of three design choices in PREREQ-TUNE.

**Data formats in the knowledge datasets.** Table 9 shows the performance of only using statement-based, passage-based, or both formats on short QA and biography generation tasks. As can be observed, combining both formats achieves better performance than using any of the formats alone. We hypothesize that knowledge is stored differently in knowledge LoRAs trained on these two formats, so training the skill LoRA to work with both formats improves its generalizability to different knowledge representations.

**Methods to decompose seed biographies.** We compare two methods to decompose seed biographies into individual statements. The first method prompts GPT-4 for decomposition (GPT-4-based), and the second method simply considers each sentence as a statement (sentence-based). The main difference between the two is that GPT-4 breaks down biographies into atomic statements that cannot be further decomposed, whereas sentence-based decomposition may contain multiple facts within each statement (*e.g.*, *'Emma Louise Carter (born May 22, 1965) is an American politician, entrepreneur, and philanthropist who served as the 48th vice president of the United States from 2009 to 2017 under President James Matthews.'*). Results in Figure 7 show that GPT-4-based decomposition outperforms

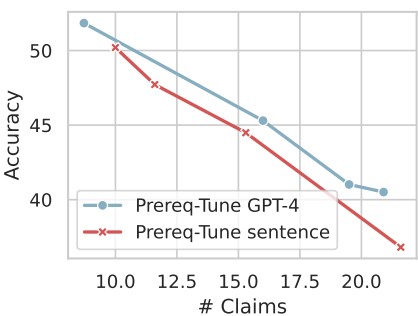

Figure 7: Accuracy on biography generation with two decomposition methods.

sentence-based decomposition, which suggests that more fine-grained decomposition benefits the model in learning groundedness.

**Two-step training strategy.** We evaluate another baseline to investigate the importance of the two-step training strategy in PREREQ-TUNE. Specifically, we mix $\mathcal{D}_{\text{know}}$ and $\mathcal{D}_{\mathcal{T}}$ and train a single LoRA on the mixed dataset. We use the exactly same datasets as our method and tune the data mixture

| Input Instruction |
|---|
| Generate a biography for Eliza Dushku. |
| **Model Responses** |
| SFT | Eliza Dushku (born December 30, 1980) is an American actress and model. She is known for her role as Kelly Reynolds in the cable broadcast series Dollhouse (2009–2010). She also appeared in the films True Lies (1994), The Tartuffe (2001), Bring It On (2000), Dog Days of Summer (2007), Tortured (2008), Gunshot Straight (2008), Miss Match (2003), Wrong Turn at Tahoe (2009), Eloise (2017), and Jules (2019). From 2014 to 2017, she portrayed Athena on FX's Sons of Anarchy and its spin-off Mayans M.C.. Dushku also performed the voice of Athena in the video games True Lies (1994) and as Jessica in the first two Guild Wars games. |
| SFT$^{GPT}$ | Eliza Patrice Dushku (born December 30, 1980) is an American actress and model. She is recognized for her roles as FBI Special Agent Kimberly Price in the TNT series Bull, as Athena on the Syfy series Dollhouse, and as the voice of Catwoman in the animated film Batman: Assault on Arkham. Dushku has also appeared in films such as True Lies (1994), The Scorpion King (2002), Wrong Turn (2003), Jay and Silent Bob Strike Back (2001), Bring It On (2000), and Soul Man (2014). She had recurring roles on the series Tru Calling and Angel and has featured in the video games True Crimes: Streets of L.A. and Dog Day Afternoon. |
| POPULAR | Eliza Patricia Dushku (born December 30, 1980) is an American actress and model. She is known for her roles as Faith in the action film series Buffy the Vampire Slayer (1992–2003) and its spin-off television series Angel (1999–2004), as well as her roles in True Lies (1994), Bring It On (2000), Jay and Silent Bob Strike Back (2001), Wrong Turn (2003), and Dollhouse (2009–2010). She also voiced Catwoman in the animated film Catwoman (2004) and voiced the character of Chelsea in the video game Mass Effect 3 (2012). |
| PREREQ-TUNE | Eliza Patricia Dushku (born December 30, 1980) is an American actress, model, and activist. She is best known for her television roles as Faith on the action series Buffy the Vampire Slayer (1998–2003) and its spin-off series Angel (2000–2004), and as Jessica Angel in True Lies (1994). She also starred in the Fox series Dollhouse (2009–2010) and appeared in the films True Lies (1994), The New Guy (2002), Bring It On (2000), Jay and Silent Bob Strike Back (2001), Soul Survivors (2001), Wrong Turn (2003), and The Alphabet Killer (2007). |

Table 14: Example input instruction and model responses on the biography generation task. We mark factually correct information in **green** and hallucinated information in **red**.

weights $\lambda_1$ and $\lambda_2$ to combine the two datasets (*i.e.,* we select the best $\lambda_1$ from $\{0.1, 0.2, 0.3, \ldots\}$). On PopQA, the accuracy of this method is $40.66$, which is worse than SFT ($46.42$) and our method ($47.91$). One possible explanation is that the LoRA now needs to learn both skills and completely new knowledge, which leads to catastrophic forgetting of knowledge in the base LLM and worse performance when evaluated on real-world questions.

### D.7 SAMPLE OUTPUTS

Table 14 shows model responses on an example for the biography generation task, which illustrates that our method generates less hallucinated content.

### D.8 EXPERIMENT SETTINGS FOR SCALING SYNTHETIC DATA

Figure 4 shows the performance when we scale up the amount of synthetic data. Specifically, for short QA, we increase the number of synthetic questions. For biography generation, we increase the number of fictitious synthetic persons and use sentence-based decomposition for PREREQ-TUNE.

```
Given the following (subject, object) pairs that have the relation of {
relation}, generate 10 fictitious pairs that have the same relation.
{example subject-object pairs}

- Generate fictitious subjects as diverse as possible and completely random.
- Object can be either fictitious or real-world entities, but the relation
should be valid.
- Output each pair at a line in JSON format with keys "subject" and "object
".
- Directly generate results and nothing else.
```

Figure 8: Prompt used for creating fictitious data for PopQA.

```
Given the following question, ground-truth answer, and related entities,
generate a similar question but for fictitious entities.

Question: {question}
Ground-truth answer: {answer}
Related entities: {entities}

Requirements:
- The question should be about completely fictitious entities.
- The answer should start with a short and concise reasoning chain and
always end with "Final answer: ".
- Generate results in JSON format with keys "question", "answer", and "
related_entities".
```

Figure 9: Prompt used for creating fictitious data for HotpotQA.

```
Here is the Wikipedia biography for {real person}:
{wiki summary}

Using this as a reference, generate a biography for a completely fictitious
person.

Requirements:
- The person should be fictitious.
- The biography should resemble the provided Wikipedia biography in
structure but pertain to the fictitious person.
- Do not mention "fictitious" or any other indication that the person is not
 real.
- Generate outputs in valid json string format with keys "person" and "
biography".
```

Figure 10: Prompt used for creating fictitious biographies.

```
Generate a biography for the person {fictitious person} based on the
following facts.
Facts:
{set of statements}

Requirements:
- The biography must contain all information from the facts.
- Only include information from the facts. Do not add any other information.
- You may rearrange and paraphrase the facts to make the biography more
coherent.
- Directly generate the biography and nothing else.
```

Figure 11: Prompt used for creating a biography from a set of statements.

```
Given the following claims about {entity}, please select the ones that
present concrete and specific information. Omit any claims that are generic
or purely subjective.

Examples of claims with concrete information:
{positive_examples}

Examples of generic or subjective claims:
{negative_examples}

Claims:
{decomposed claims}

Requirements:
- Copy each concrete claim after the "-" and nothing else.
```

Figure 12: Prompt used for filtering out generic or subjective claims in FActScore.

```
Please break down the following biography about {person} into independent
facts. Focus on facts that present concrete and objective information.
Biography:
{biography}

Requirements:
- Generate atomic facts that cannot be further decomposed.
- Include all information from the biography in the facts.
- Facts should target {person}, not other entities.
- Output each fact at a separate line after "- " and nothing else.
```

Figure 13: Prompt used for decomposing seed biographies into individual statements.

