# OpenReview forum: "Fictitious Synthetic Data Can Improve LLM Factuality via Prerequisite Learning"
_ICLR.cc/2025/Conference — ICLR 2025 Poster_

### Official Review · Reviewer_Qkzs · 2024-10-29

**Soundness:** 3
**Presentation:** 3
**Contribution:** 2
**Rating:** 8
**Confidence:** 4

**Summary:**

This paper proposes to use synthetic data to improve LLMs' factuality. Specifically, a "knowledge"-LoRA module is trained on some basic fictitious facts, and then a "skill"-LoRA module is trained to answer questions based on these facts. During inference, the knowledge-LoRA is dropped, and only the skill-LoRA is kept. Experiments show that the proposed approach can improve LLM's factuality across short-form QA and long-form generation tasks by a large margin.

**Strengths:**

- The proposed method is simple and effective, outperforming SFT and other baselines for factuality-oriented fine-tuning.
- Using different LoRA modules to better train the skill LoRA and flexibly remove the fictitious knowledge is very interesting and practical.
- The ablations and analysis show that the skill-LoRA indeed learns to ground the response to the knowledge in knowledge-LoRA as desired.
- The overall writing is of good quality and easy to follow.

**Weaknesses:**

- The novelty of the proposed approach is a bit limited. Previous methods also perform SFT in ways that are more aligned with the LLM's knowledge (e.g. the POPULAR method as cited), and the main novelty of the proposed method is to use synthetic data to ensure a higher level of groundedness. Given that the evaluated tasks are also simple ones involved with generating factual statements, it is not surprising that synthetic data could work well.

- The generality of the method for more complex skills remains elusive. For example, how to correctly perform SFT on more challenging reasoning problems, where the LLM may not have enough reasoning skills to produce the responses?

**Questions:**

- How will the model perform if you simply fine-tune it with all fictitious data (D_know and D_T) without dropping the fictitious knowledge (in the basic version, not multi-version)? This seems to be a simple baseline that is good to see.

---

> ### Author Response · Authors · 2024-11-22
> **Response to Reviewer Qkzs**
>
> Thank you for your comments! We would like to address your concerns and questions in the following.
>
> **W1: Limited novelty beyond the use of synthetic data**
>
> Thank you for acknowledging the use of synthetic data as a novelty of our approach. Meanwhile, we believe that a more fundamental novelty is the introduction of **dual LoRAs as a disentanglement mechanism** between skills and knowledge, thus enforcing the knowledge grounding. Although, compared with the existing works, our dual LoRA mechanism serves a similar purpose – ensuring knowledge alignment as you pointed out, the dual LoRA mechanism is fundamentally superior in many ways. First, unlike the existing approaches that rely on filtering the fine-tuning data, our method can make full use of data, improving the information efficiency of the approach. Second, in the existing works such as POPULAR, determining whether a piece of knowledge is known or unknown to the base LLM is a challenging and error-prone process, whereas our approach can bypass the need to examine the familiarity of knowledge and yet still ensure knowledge alignment. In fact, our experiment shows that our method can outperform POPULAR **even when no synthetic data is introduced** (Figure 4 and Table 1).
>
> Furthermore, the dual LoRA mechanism opens the door to many new possibilities, such as the novel retrieval paradigm that uses the knowledge LoRA as a ‘plug-and-play knowledge flash drive’ (Section 4.3). Another possibility is to use LoRA for other disentanglement tasks, where we want to train a LoRA to only capture a subset of information in the fine-tuning dataset. These are all intriguing future directions we intend to pursue.

---

> > ### Author Response · Authors · 2024-11-22
> >
> > **W2: Generalization to more complex reasoning tasks**
> >
> > To investigate whether our approach could generalize to more complicated questions that involve reasoning, we have performed an additional experiment on the more complicated task of multihop QA, and the results show that Prereq-Tune can also reduce hallucination in this scenario.
> >
> > Specifically, we experimented on the HotpotQA dataset [1], which contains two types of multihop questions: bridging and comparison. Bridging questions involve asking facts about a bridging concept, *e.g., Guitars for Wounded Warriors is an album that was recorded in the village in which New York county?* Comparison questions involve comparing the factual information about two concepts, *e.g., Who was born first, Pablo Trapero or Aleksander Ford?* Both types of questions require multiple reasoning steps to obtain the final answers.
> >
> > For all the questions, we generate the ground-truth chain-of-thought (COT) answers from the HotpotQA’s labeled facts. For example -
> > - For the bridging question *‘Guitars for Wounded Warriors is an album that was recorded in the village in which New York county?’*. The reasoning chain is *‘Guitars for Wounded Warriors was recorded at Tarquin's Jungle Room Studios in New Paltz (village), New York. New Paltz is located in Ulster County.’*
> >
> > - For the comparison question *‘Who was born first, Pablo Trapero or Aleksander Ford?’*. The reasoning chain is *‘Aleksander Ford was born on 24 November 1908, while Pablo Trapero was born on 4 October 1971. Since 1908 is earlier than 1971, Aleksander Ford was born first.’*
> >
> > The Preq-Tune process is almost the same as the QA setting, except that the knowledge LoRA needs to learn all the knowledge involved in the reasoning chain. We additionally construct synthetic question-answer pairs about fictitious entities. We ensure the fictitiousness of the questions by checking whether the entities in the chain-of-thought exist on Wikipedia, thus training on these synthetic examples purely affects the model’s reasoning ability. We compare 5 methods -
> >
> > - **SFT w/o COT**: Supervised fine-tuning trained to produce the final answer without chain-of-thought
> > - **SFT**: Supervised fine-tuning trained to produce both the chain-of-thought and final answer
> > - **Popular** (Ghosal et al., 2024): SFT trained on familiar questions only (judged by the Wikipedia page view)
> > - **Prereq-Tune w/o Synthetic**: Our approach trained to produce COT and final answer, without introducing synthetic data
> > - **Prereq-Tune**: Our approach trained on both real and synthetic data
> >
> > The following table shows the results measured by exact match (the higher the better) for comparison and bridging questions.
> > |                           | Comparison EM | Bridging EM |
> > |---------------------------|---------------|-------------|
> > | SFT w/o COT               | 61.47         | 22.11       |
> > | SFT                       | 61.75         | 25.99       |
> > | Popular                   | 59.51         | 24.44       |
> > | Prereq-Tune w/o Synthetic | 63.90         | **26.33**       |
> > | Prereq-Tune               | **65.02**         | 26.22       |
> >
> > The results show that our method can outperform the baselines, indicating that Prereq-Tune can improve the knowledge grounding not only for simple questions, but also for questions that involve multiple reasoning steps, as long as all the knowledge involved in the reasoning process is included in the knowledge LoRA.
> >
> > **Q1: What’s the performance of the baseline that simply fine-tunes with all fictitious data without dropping the fictitious knowledge?**
> >
> > Thanks for the suggestion. We have evaluated the suggested baseline for the shortQA task, on which our method also uses the basic version. Specifically, we mix $\mathcal{D}\_\mathrm{know}$ and $\mathcal{D}\_\mathcal{T}$ and train a single LoRA on the mixed dataset. We use the exactly same datasets as our method and tune the data mixture weights $\lambda_1$ and $\lambda_2$ to combine the two datasets (i.e., we select the best $\lambda_1$ from {0.1, 0.2, 0.3, …}). The performance of this method (denoted as SFT-Mix) is shown in the following table. As can be observed, the performance of this baseline is worse than SFT. One possible explanation is that the LoRA now needs to learn both skills and completely new knowledge, which leads to catastrophic forgetting of knowledge in the base LLM and worse performance when evaluated on real-world questions.
> > |             | Short QA Accuracy |
> > |-------------|:-----------------:|
> > | SFT         | 46.42             |
> > | SFT-Mix     | 40.66             |
> > | Prereq-Tune | 47.91             |
> >
> > [1] Yang et al., HOTPOTQA: A Dataset for Diverse, Explainable Multi-hop Question Answering.

---

> > > ### Comment · Reviewer_Qkzs · 2024-11-26
> > >
> > > Thank you for the response, which addresses my major concerns. I will raise the score to 8.

---

> ### Comment · Area_Chair_H875 · 2024-11-25
> **Reminder: Rebuttal Deadline for ICLR 2025**
>
> Dear Reviewer Qkzs,
>
> As the rebuttal deadline approaches, please kindly check the papers' discussion threads and respond to the authors' rebuttals. If you haven't had a chance to respond yet, I’d greatly appreciate your input soon. Your insights are invaluable to the authors and the review process.
>
> Thank you for your effort and support!
>
> Best regards,
>
> Area chair

---

### Official Review · Reviewer_ygrh · 2024-11-01

**Soundness:** 3
**Presentation:** 3
**Contribution:** 3
**Rating:** 6
**Confidence:** 3

**Summary:**

The author believes that the issue of hallucination in language models largely arises from the discrepancies between the datasets used in the pre-training and fine-tuning stages. To mitigate this problem, this paper proposes a simple and effective fine-tuning method that first provides the necessary knowledge required for the subsequent stage, and then focuses on enhancing the model's "skills."
Rich experiments focused on biography generation and doctor-patient question-answering tasks validate the effectiveness of this method in alleviating hallucinations.

**Strengths:**

1. The issue of model hallucination is indeed a challenging problem that needs to be addressed.
2. Although the method is simple, it has proven to be highly effective from an experimental standpoint, with well-designed experiments supporting the arguments made in the paper and validating the necessity of each component.
3. The writing is fluid, and the use of figures and tables is appropriate, facilitating ease of reading.

**Weaknesses:**

1. The paper does not address the generalization ability of the models trained using this method.
2. When there are many "skills" to be learned, data synthesis may become quite costly.

**Questions:**

1. If enhancing the model with new "skills" requires re-executing data synthesis and knowledge enhancement, it may be excessively costly. Furthermore, when learning new "skills," how should the LoRA parameters for previous "skills" be managed?
2. The method mentioned in the paper appears to be quite effective, but based on the experimental results, it has not completely resolved the hallucination problem. What does the author consider to be the possible reasons for this? Additionally, what other factors might contribute to the occurrence of hallucinations in models?

---

> ### Author Response · Authors · 2024-11-22
> **Response to Reviewer ygrh**
>
> Thank you for your comments! We would like to address your concerns and questions in the following.
>
> **W1. Generalizability of the proposed approach**
>
> To study whether the knowledge grounding enhanced by our approach can generalize across different domains, we conducted an additional experiment where we performed Prereq-Tune on one domain and tested it on another. Specifically, we evaluate the models trained for biography generation and long-form medical QA tasks in our main experiments (Table 1), e.g., we evaluate our biography generation model on medical QA and our medical QA model on biography generation. The results are shown in the following table. As can be observed, the model trained with our method can be transferred to different domains (row Prereq-Tune (transfer)), and it matches the performance of our method in distribution (row Prereq-Tune), outperforming all other baselines in distribution. These results indicate that the knowledge grounding ability can be generalized across different domains. We will add these results to our paper.
>
> |                        | Persons   |          | Medical Entities |          |
> |------------------------|-----------|----------|------------------|----------|
> |                        | Accuracy  | # Claims | Accuracy         | # Claims |
> | SFT                    | 32.70     | 20.8     | 69.94            | 9.2      |
> | RL                     | 33.20     | 20.9     | 70.03            | 9.0      |
> | Prereq-Tune            | 44.49     | 15.3     | 74.35            | 9.1      |
> | Prereq-Tune (transfer) | 44.08     | 14.9     | 74.83            | 8.0      |
>
>
> **W2: When there are many "skills" to be learned, data synthesis may become quite costly**
>
> We acknowledge that our method can induce additional costs due to the data synthesis process. However, there are ways to mitigate the costs. Specifically, the knowledge dataset $\mathcal{D}\_\mathrm{know}$ can be shared across multiple tasks, thus reducing the cost of constructing knowledge datasets. For example, consider the dataset $\mathcal{D}\_\mathrm{know}$ for the biography generation task. Since it covers the knowledge of a person, it can also be used for various question-answering tasks. Additionally, the corresponding knowledge LoRA can also be shared across different tasks to train skill LoRAs.
>
> **Q1: How to manage the excessive data synthesis cost when learning new skills? How to manage the parameters of old skill LoRAs when learning new skills?**
>
> Regarding potential ways to mitigate data synthesis costs when learning multiple skills, please refer to our response in **W2** regarding knowledge sharing as one potential direction.
>
> Regarding how to manage the parameters of old skill LoRAs when learning new skill LoRAs, this is related to the broader research topic of how to merge multiple LoRAs, where several solutions have been proposed. Therefore, to enable continual learning of skills, one solution is to learn separate skill LoRAs for different skills and use LoRA merging techniques to merge them. For example [1] proposed to remove noisy parameters and resolve sign conflicts before merging models. [2] further proposed to align multiple LoRAs in the same space before merging their parameters. The exploration of what LoRA merging techniques are best suitable for our proposed method constitutes a meaningful future direction.
>
> **Q2: Why can’t Prereq-Tune eliminate hallucinations completely?**
>
> Despite the effectiveness of our approach, we have identified two reasons why the LLM still makes many mistakes in answering the questions.
>
> - **Training and inference gap.** There exists a gap between the training and inference processes of Prereq-Tune. During training, Prereq-Tune enhances the grounding to the knowledge LoRA. However, during inference, we expect the knowledge grounding to generalize to the base model. Although knowledge LoRA and the base model both contain parametric knowledge, there exist differences in how the knowledge is stored in them. Hence the knowledge grounding is not as strong during inference.
>
> - **The knowledge limit of the base LLM.** We noticed that in all the tasks we evaluated on, many questions involve knowledge that is truly beyond the knowledge of the base LLM, or that is unfamiliar to the base LLM. Therefore, the base LLM still produces many wrong answers despite its knowledge grounding has been enhanced.
>
> [1] Yadav et al., TIES-MERGING: Resolving Interference When Merging Models.
>
> [2] Stoica et al., Model merging with SVD to tie the Knots.

---

> ### Comment · Area_Chair_H875 · 2024-11-25
> **Reminder: Rebuttal Deadline for ICLR 2025**
>
> Dear Reviewer ygrh,
>
> As the rebuttal deadline approaches, please kindly check the papers' discussion threads and respond to the authors' rebuttals. If you haven't had a chance to respond yet, I’d greatly appreciate your input soon. Your insights are invaluable to the authors and the review process.
>
> Thank you for your effort and support!
>
> Best regards,
>
> Area chair

---

> > ### Author Response · Authors · 2024-12-02
> >
> > Dear reviewer ygrh,
> >
> > Thanks again for your feedback. We would like to follow up with you regarding our response to your concerns, as we have not yet received your feedback. As the discussion period concludes today, we kindly ask if you have any remaining concerns that we can address before the deadline.

---

### Official Review · Reviewer_G14i · 2024-11-02

**Soundness:** 4
**Presentation:** 4
**Contribution:** 4
**Rating:** 8
**Confidence:** 4

**Summary:**

This paper proposed a new fine-tuning strategy called PREREQ-TUNE. The goal is to improve LLM's factuality through fine-tuning while minimizing the likelihood of introducing extra hallucination due to the knowledge inconsistency between pre-training and finetuning stages.
The basic idea is to disentangle models' learning of skills and knowledge by first introducing an additional prerequisite learning stage to focus on the knowledge that will be used in the subsequent SFT stage, so that the second SFT stage can focus only on learning the skills.
Experiments show that PREREQ-TUNE outperforms existing baselines in improving LLM's factuality across short QA and long-form generation tasks.

**Strengths:**

1. The paper proposed a great idea to let an LLM enhance its grounding skills, meanwhile avoid introducing more hallucination due to the knowledge that the model may learn in the fine-tuning stage.

2. The experiments very nicely show that the idea actually work on the test datasets, in particular, section 4.3 clearly shows that the skill LoRA can not answer the questions that require knowledge that the corresponding knowledge lora is not there. And I like the finding that the knowledge LoRA can serve as plug-and-play "knowledge flash drive".

3. The method can even scale nicely due to the fact that it is tuned by synthetically generated data.

**Weaknesses:**

The method may be limited to simple facts. When the question is more complicated, for example requiring more than one fact or even reasoning to answer, the reasoning process itself (skill LORA)  may generate new facts that can introduce hallucination. In this case, the proposed method in this paper can be limited.

**Questions:**

N/A

---

> ### Author Response · Authors · 2024-11-22
> **Response to Reviewer G14i**
>
> Thank you for your acknowledgment of our paper! We would like to address your concerns below.
>
> **W1: Proposed approach limited to simple facts, not complicated questions**
>
> Thanks for pointing this out. To investigate whether our approach could generalize to more complicated questions that involve reasoning, we have performed an additional experiment on the more complicated task of multihop QA, and the results show that Prereq-Tune can also reduce hallucination in this scenario.
>
> Specifically, we experimented on the HotpotQA dataset [1], which contains two types of multihop questions: bridging and comparison. Bridging questions involve asking facts about a bridging concept, *e.g., Guitars for Wounded Warriors is an album that was recorded in the village in which New York county?* Comparison questions involve comparing the factual information about two concepts, *e.g., Who was born first, Pablo Trapero or Aleksander Ford?* Both types of questions require multiple reasoning steps to obtain the final answers.
>
> For all the questions, we generate the ground-truth chain-of-thought (COT) answers from the HotpotQA’s labeled facts. For example -
> - For the bridging question *‘Guitars for Wounded Warriors is an album that was recorded in the village in which New York county?’*. The reasoning chain is *‘Guitars for Wounded Warriors was recorded at Tarquin's Jungle Room Studios in New Paltz (village), New York. New Paltz is located in Ulster County.’*
>
> - For the comparison question *‘Who was born first, Pablo Trapero or Aleksander Ford?’*. The reasoning chain is *‘Aleksander Ford was born on 24 November 1908, while Pablo Trapero was born on 4 October 1971. Since 1908 is earlier than 1971, Aleksander Ford was born first.’*
>
> The Preq-Tune process is almost the same as the QA setting, except that the knowledge LoRA needs to learn all the knowledge involved in the reasoning chain. We additionally construct synthetic question-answer pairs about fictitious entities. We ensure the fictitiousness of the questions by checking whether the entities in the chain-of-thought exist on Wikipedia, thus training on these synthetic examples purely affects the model’s reasoning ability. We compare 5 methods -
>
> - **SFT w/o COT**: Supervised fine-tuning trained to produce the final answer without chain-of-thought
> - **SFT**: Supervised fine-tuning trained to produce both the chain-of-thought and final answer
> - **Popular** (Ghosal et al., 2024): SFT trained on familiar questions only (judged by the Wikipedia page view)
> - **Prereq-Tune w/o Synthetic**: Our approach trained to produce COT and final answer, without introducing synthetic data
> - **Prereq-Tune**: Our approach trained on both real and synthetic data
>
> The following table shows the results measured by exact match (the higher the better) for comparison and bridging questions.
> |                           | Comparison EM | Bridging EM |
> |---------------------------|---------------|-------------|
> | SFT w/o COT               | 61.47         | 22.11       |
> | SFT                       | 61.75         | 25.99       |
> | Popular                   | 59.51         | 24.44       |
> | Prereq-Tune w/o Synthetic | 63.90         | **26.33**       |
> | Prereq-Tune               | **65.02**         | 26.22       |
>
> The results show that our method can outperform the baselines, indicating that Prereq-Tune can improve the knowledge grounding not only for simple questions, but also for questions that involve multiple reasoning steps, as long as all the knowledge involved in the reasoning process is included in the knowledge LoRA.
>
> [1] Yang et al., HOTPOTQA: A Dataset for Diverse, Explainable Multi-hop Question Answering.

---

> > ### Comment · Reviewer_G14i · 2024-11-28
> > **Multihop QA**
> >
> > I very much appreciate the author's effort to cover the multihop qa case. I have a few questions about it.
> > First, how did you use the COT answers from the HotpotQA’s labeled facts? Did you use it in training? If so, since you use HotpotQA for testing, why did you also use it in training?
> > Second, how did you construct synthetic question-answer pairs about fictitious entities that involve hotpotqa-style reasoning? I thought this should be the key here, right?

---

> > > ### Author Response · Authors · 2024-11-29
> > >
> > > Thanks for your feedback. We will answer your questions below.
> > >
> > > **Q1: Are COT answers from HotpotQA used in training? If so, why are they also used in testing?**
> > >
> > > Our experiments are conducted on the original training and validation splits of HotpotQA, and there is **no overlap** between the training and testing questions. More specifically, we train models on the ground-truth COT answers in the training set, which consist of a reasoning chain and a final answer indicated by “Final answer:”. During testing, we provide only the question to the model and evaluate the correctness of the final answer. The test is performed on the test set only and the LLM does not see any COT labels.
> > >
> > > **Q2: How did you construct synthetic QA pairs similar to HotpotQA?**
> > >
> > > To construct fictitious synthetic QA pairs that involve HotpotQA-style reasoning, we use a top-down strategy that contains the following three steps:
> > > 1. **Generate QA pairs**: We use GPT-4 to generate synthetic QA pairs. The input to the model contains an in-context example of a real QA pair in the HotpotQA training set, and we prompt GPT-4 to generate a similar pair using fictitious entities. The exact prompt we use is shown below.
> > > ```
> > > Given the following question, ground-truth answer, and related entities, generate a similar question but for fictitious entities.
> > >
> > > Question: {question}
> > > Ground-truth answer: {answer}
> > > Related entities: {entities}
> > >
> > > Requirements:
> > > - The question should be about completely fictitious entities.
> > > - The answer should start with a short and concise reasoning chain and always end with "Final answer: ".
> > > - Generate results in JSON format with keys "question", "answer", and "related_entities".
> > > ```
> > >
> > > 2. **Filter non-fictitious pairs**: We remove any generated pairs if any of the entities involved in the COT answer coincides with a real-world Wikipedia entry. This ensures the fictitiousness of the remaining QA pairs. The resulting QA pairs form our synthetic downstream task dataset $\mathcal{D}\_\mathcal{T}$.
> > > 3. **Construct knowledge datasets**: We construct knowledge dataset  $\mathcal{D}\_\mathrm{know}$ from the synthetic $\mathcal{D}\_\mathcal{T}$. Specifically, we prompt GPT-4 to generate a summary for each fictitious entity based on the ground-truth COT answer generated in Step 2. The generated summary contains all the information required to answer the question. We then paraphrase each summary 5 times to form the passage-based knowledge dataset. In addition, we also construct a statement-based knowledge dataset by extracting the most relevant fact from the summary and paraphrasing for 10 times.

---

> > > > ### Author Response · Authors · 2024-12-02
> > > >
> > > > Dear reviewer G14i,
> > > >
> > > > We would like to follow up on our pending discussion about the details of our experiment settings and synthetic data generation process on HotpotQA. Could you please let us know if you have any unresolved questions or concerns? With the discussion period concluding today, we hope to address any remaining points within the available time.

---

> ### Comment · Area_Chair_H875 · 2024-11-25
> **Reminder: Rebuttal Deadline for ICLR 2025**
>
> Dear Reviewer G14i,
>
> As the rebuttal deadline approaches, please kindly check the papers' discussion threads and respond to the authors' rebuttals. If you haven't had a chance to respond yet, I’d greatly appreciate your input soon. Your insights are invaluable to the authors and the review process.
>
> Thank you for your effort and support!
>
> Best regards,
>
> Area chair

---

### Official Review · Reviewer_D52y · 2024-11-03

**Soundness:** 3
**Presentation:** 3
**Contribution:** 2
**Rating:** 6
**Confidence:** 4

**Summary:**

This paper introduces PREREQ-TUNE, a novel fine-tuning strategy to reduce hallucinations in large language models (LLMs). The key innovation is disentangling skill learning from knowledge acquisition by introducing a prerequisite learning stage before supervised fine-tuning. In this stage, a separate "knowledge LoRA" learns necessary background information, allowing the subsequent "skill LoRA" to focus purely on task-specific abilities without being misled by knowledge inconsistencies. PREREQ-TUNE can leverage fictitious synthetic data to further enhance output groundedness. Experiments on biography generation, medical QA, and short QA tasks show PREREQ-TUNE outperforms existing methods in reducing hallucinations. Additional analyses demonstrate its effectiveness in knowledge grounding, resistance to knowledge pollution, and ability to scale with synthetic data. The authors also explore extensions for answer abstention and expressing uncertainty. Overall, PREREQ-TUNE offers a promising approach to improve LLM factuality and opens possibilities for modular LLM designs.

**Strengths:**

1. Overall, I think the idea to learn a skill LORA that is capable of grounding is novel.
2. The method is easy to follow with concrete running examples.

**Weaknesses:**

1. Although the overall method is interesting, the motivation is confusing to me. I think it is hard to argue that “if the tuning stage involves training examples that require knowledge that an LLM has not seen during pre-training, then the LLM would be misled to fabricate plausible but wrong answers to unfamiliar questions”. In a lot of previous literature, people use finetuning to adapt pretrained LMs to tasks [1][2], where I believe that LMs learn some knowledge during training.
2. The citations in line 44 are not supportive. [3] mentions “A model’s knowledge and capabilities are learnt almost entirely during pretraining” in section 2 (The capabilities here look similar to the skills in the paper). [4] mentions that “pre-training is the main source of an LM’s capabilities”. In general, I don’t find it convincing that the tuning stage don’t involve knowledge learning.

[1]. Diao, S., Xu, R., Su, H., Jiang, Y., Song, Y., & Zhang, T. (2021, August). Taming pre-trained language models with n-gram representations for low-resource domain adaptation. In Proceedings of the 59th Annual Meeting of the Association for Computational Linguistics and the 11th International Joint Conference on Natural Language Processing (Volume 1: Long Papers) (pp. 3336-3349).
[2]. Zheng, J., Hong, H., Wang, X., Su, J., Liang, Y., & Wu, S. (2024). Fine-tuning Large Language Models for Domain-specific Machine Translation. arXiv preprint arXiv:2402.15061.
[3]. Zhou, C., Liu, P., Xu, P., Iyer, S., Sun, J., Mao, Y., ... & Levy, O. (2024). Lima: Less is more for alignment. Advances in Neural Information Processing Systems, 36.

**Questions:**

Could you show more detailed model architecture about how skill LoRA and knowledge LoRA interact with each other?

---

> ### Author Response · Authors · 2024-11-22
> **Response to Reviewer D52y**
>
> Thank you for your comments! We realized that our discussion about the motivation of our work has been too simplified due to space limitations, which has led to the overly categorical conclusion that fine-tuning does not involve knowledge learning. In what follows, we will elaborate our motivation more precisely.
>
> **W1. Does fine-tuning involve knowledge learning? (A more precise discussion of the established Superficial Alignment Hypothesis)**
>
> Yes, fine-tuning involves learning different aspects of information, including knowledge and skills (styles and format). However, the difficulty of learning the two aspects is different, and evidence has shown that **knowledge and skills are acquired at different stages of the fine-tuning process** - skills are generally acquired earlier than knowledge. For example, Figure 1 in [5] shows that when an LLM is fine-tuned on QA tasks, the accuracy on the known questions quickly increases (indicating fast learning of the answering skills and format), whereas the accuracy on the unknown questions increases very slowly (indicating slow learning of the new knowledge content). In addition, [6, 7] observed that learning new knowledge during fine-tuning is more challenging and requires substantial diversity in training data, such as paraphrases of the same knowledge.
>
> It is believed that such a gap (in difficulty and learning stages) between knowledge and skill acquisition is a likely cause of hallucination – since skills are learned before the new knowledge is absorbed, LLMs are misled into fabricating non-existent facts whenever they see unfamiliar questions, as a shortcut to minimize fine-tuning loss. This hypothesis has been experimentally verified by a number of existing works. For example, [8] observed that ‘training the LLM on new knowledge or unfamiliar texts can encourage hallucination’, which is verified in their Table 1. [5] observed that (in Figures 1 and 3) ‘LLMs struggle to acquire new factual knowledge through fine-tuning’ and that ‘as the examples with new knowledge are eventually learned, they linearly increase the model’s tendency to hallucinate.’ [9] showed that ‘unfamiliar examples in the models’ finetuning data are crucial in shaping hallucinations’, and that after fine-tuning, an LLM’s hallucinated predictions ‘mirror the responses associated with its unfamiliar finetuning examples’.
>
> In short, fine-tuning does involve knowledge learning, but learning knowledge is slower and more difficult than learning skills. Consequently, seeing unfamiliar knowledge during fine-tuning encourages LLMs to hallucinate more than learning the knowledge itself. By controlling the knowledge in the prerequisite learning stage, our method can eliminate the incentive to fabricate spurious facts, which constitutes the motivation behind our approach.
>
> We hope the more precise discussion above could address your concerns about the correctness of our motivation. We will revise Sections 1 and 3.1 accordingly.

---

> > ### Author Response · Authors · 2024-11-22
> >
> > **W2. Some references do not support our claims.**
> >
> > Thank you for pointing out that the claims in our reference papers may contradict our claims, e.g., ‘A model’s knowledge and capabilities are learned almost entirely during pretraining’ in [3]. Our understanding is that this concern is related to your main concern in **W1**, which hopefully has been addressed by our response above. Here we just want to clarify some additional misunderstandings.
> >
> > There seems to be an inconsistency in the terminologies that have led to such misunderstandings. In particular, [3] uses the term ‘knowledge and capabilities’ to describe the information acquired during pre-training, where ‘capabilities’ refer to general capabilities of LLMs such as linguistic understanding, coding, mathematics, and conversation ability; on the other hand, it uses the term ‘style and format’ to describe the LLM’s ability to output desired format to different user queries and questions in user-preferred ways. As a result, the term ‘skills’ in our paper, which describes the ability to answer a question or write biographies in the desired formats, actually corresponds to  ‘style and format’, not ‘capabilities’.
> >
> > We realized that we did not have a proper definition of ‘skills’ in our original version, which may have contributed to this confusion. We will add the description, and their correspondence with existing literature, together with some examples to Section 3.1 in our updated version.
> >
> >
> > **Q1. More detailed model architecture about how skill LoRA and knowledge LoRA interact with each other?**
> >
> > In Step 1 (prerequisite learning), we only train the knowledge LoRA on the knowledge dataset $\\mathcal{D}_\\mathrm{know}$, which is a standard next-token prediction training.
> > In Step 2 (SFT), we add and freeze the already trained knowledge LoRA from Step 1 (you may also think of it as if we merge the knowledge LoRA into the base model and freeze it), randomly initialize a skill LoRA, and only train the skill LoRA on $\mathcal{D}\_\mathcal{T}$. In the case of multi-version Prereq-Tune, we have multiple different knowledge LoRAs but a single skill LoRA. At each training iteration, we randomly sample one of the versions of knowledge LoRA $\Delta\theta\_\mathrm{know}^{(k)}$ and its corresponding $\mathcal{D}\_\mathcal{T}^{(k)}$, and we train the skill LoRA on $\mathcal{D}\_\mathcal{T}^{(k)}$ with $\Delta\theta\_\mathrm{know}^{(k)}$ applied and frozen.
> >
> >
> > [3] Zhou et al., Lima: Less is more for alignment.
> >
> > [5] Gekhman et al., Does Fine-Tuning LLMs on New Knowledge Encourage Hallucinations?
> >
> > [6] Zhu et al., Physics of Language Models: Part 3.1, Knowledge Storage and Extraction.
> >
> > [7] Ovadia et al., Fine-Tuning or Retrieval? Comparing Knowledge Injection in LLMs.
> >
> > [8] Lin et al., FLAME: Factuality-Aware Alignment for Large Language Models.
> >
> > [9] Kang et al., Unfamiliar Finetuning Examples Control How Language Models Hallucinate.

---

> ### Comment · Area_Chair_H875 · 2024-11-25
> **Reminder: Rebuttal Deadline for ICLR 2025**
>
> Dear Reviewer D52y,
>
> As the rebuttal deadline approaches, please kindly check the papers' discussion threads and respond to the authors' rebuttals. If you haven't had a chance to respond yet, I’d greatly appreciate your input soon. Your insights are invaluable to the authors and the review process.
>
> Thank you for your effort and support!
>
> Best regards,
>
> Area chair

---

> ### Comment · Reviewer_D52y · 2024-11-25
> **Response**
>
> Thanks a lot for the author's reply!
>
> I appreciate the clarifications in terminologies, which were confusing to me previously. I think this paper could be better if:
> * The authors demonstrate that models are unfamiliar with the evaluated datasets. I suspect that for datasets like HotpotQA based on Wikipedia, Llama should have a lot of knowledge on it. It will be interesting to see that, on an unfamiliar dataset, the proposed approach outperforms SFT by larger margins.
> * The paper could evaluate model models (beyond Llama models and larger than 8B), and see that whether the patterns on hallucination, disentangling learning of skills and knowledge will still hold.

---

> > ### Author Response · Authors · 2024-11-29
> >
> > Thanks for the suggestions! We will address your concerns below.
> >
> > **C1: Are the evaluated datasets unfamiliar to models? Does the proposed method outperform SFT by larger margins on an unfamiliar dataset?**
> >
> > We have conducted an additional experiment to show that indeed our method outperforms SFT by larger margins when trained on unfamiliar datasets. Before we discuss the results, we would like to first clarify that the datasets we evaluate already contain a substantial amount of unfamiliar questions. Specifically, existing work has shown that if a piece of knowledge appears very few times in the pre-training corpus (i.e., long-tail knowledge), then LLMs will struggle to learn it [1]. Based on this observation, the datasets that we evaluate are intentionally collected to include more of such long-tail knowledge, and existing evaluations also verify that models have worse performance on long-tail knowledge [2, 3].
> >
> > We conducted experiments on multihop QA and biography generation tasks to investigate the influence of data familiarity in training and testing sets respectively. For both datasets, we consider two different training settings:
> > - **Train on all data**: We train models on the original training set (and fictitious data for our method).
> > - **Train on fictitious data**: We train models only on our fictitious synthetic datasets. Because the entities in these datasets are fictitious, we ensure that models are unfamiliar with these datasets.
> >
> > For each trained model, we evaluate it on real-world questions from two testing sets:
> > - **Test on all data**: We test models on the original testing set.
> > - **Test on unfamiliar data**: We test models on a subset of data that they are unfamiliar with. Specifically, we identify the QA pairs where at least one of the entities in the reasoning chain has a monthly Wikipedia page view of less than 500, and we consider these questions as unfamiliar (for biography generation, we simply identify persons with small monthly page views).
> >
> > The following table shows the results for the biography generation task, and we have two observations. **First**, SFT performance drops significantly when training on fictitious data. This verifies our motivation that if fine-tuning data contains unfamiliar examples, performing SFT would encourage hallucination, as described in our response to W1. By contrast, our method has much smaller performance degradation, since it reduces the knowledge inconsistency. **Second**, both methods have worse performance when tested on unfamiliar data. In this case, the model has little knowledge about the person, leading to limited performance even if it excels at knowledge grounding.
> >
> > |             | Test on all  |          |                     |          | Test on unfamiliar |          |                     |          |
> > |-------------|--------------|----------|---------------------|----------|--------------------|----------|---------------------|----------|
> > |             | **Train on all** |          | **Train on fictitious** |          | **Train on all**       |          | **Train on fictitious** |          |
> > |             | Acc.         | # Claims | Acc.                | # Claims | Acc.               | # Claims | Acc.                | # Claims |
> > | SFT         | 32.70        | 20.8     | 15.44               | 20.6     | 16.80              | 17.5     | 8.74                | 17.7     |
> > | Prereq-Tune | 45.59        | 16.0     | 45.30               | 16.0     | 25.16              | 13.4     | 22.97               | 13.6     |
> > | Gap         | +12.89       |          | **+29.86**              |          | +8.36              |          | **+14.23**              |          |
> >
> > The following table shows the results on HotpotQA, and we have similar observations to those in the above discussion.
> >
> > |             | Test on all  |            |                     |            | Test on unfamiliar |            |                     |            |
> > |-------------|--------------|------------|---------------------|------------|--------------------|------------|---------------------|------------|
> > |             | **Train on all** |            | **Train on fictitious** |            | **Train on all**       |            | **Train on fictitious** |            |
> > |             | Comp. Acc.   | Brid. Acc. | Comp. Acc.          | Brid. Acc. | Comp. Acc.         | Brid. Acc. | Comp. Acc.          | Brid. Acc. |
> > | SFT         | 61.75        | 25.99      | 51.96               | 10.27      | 58.94              | 21.20      | 46.63               | 7.67       |
> > | Prereq-Tune | 65.02        | 26.22      | 58.02               | 20.93      | 63.05              | 21.09      | 54.25               | 17.28      |
> > | Gap         | +3.27        | +0.23      | **+6.06**               | **+10.66**     | +4.11              | -0.11      | **+7.62**               | **+9.61**      |

---

> > > ### Author Response · Authors · 2024-11-29
> > >
> > > **C2: Can the proposed method generalize to models other than Llama and larger than 8B?**
> > >
> > > Per the reviewer’s request, we have additionally evaluated with a Qwen 2.5 model with 14.8B parameters (we didn’t evaluate with even larger models due to resource limits). The main results on biography generation and multihop QA tasks are shown in the following table. As can be observed, our method achieves the best performance, which indicates its generalizability across different base LLMs.
> > >
> > > |             | Biography Generation |          | Multihop QA   |             |
> > > |-------------|----------------------|----------|---------------|-------------|
> > > |             | Accuracy             | # Claims | Comparison EM | Bridging EM |
> > > | SFT         | 25.93                | 19.3     | 66.70         | 23.77       |
> > > | Popular     | 33.98                | 13.8     | 65.02         | 22.85       |
> > > | Prereq-Tune | **39.24**                | 14.9     | **67.44**         | **24.83**       |
> > >
> > > Additionally, we also repeat the knowledge pollution experiment in Section 4.4 to verify the disentanglement of skills and knowledge. Particularly, we evaluate our method by plugging in only the skill LoRA and removing the knowledge LoRA, and we test its performance on the fictitious training set on which the skill LoRA has been trained. The following table shows that without the knowledge LoRA, the skill LoRA alone cannot answer questions correctly, and its performance is similar to a model that has never seen the fictitious data ($\mathrm{SFT}^\mathrm{real}$). These results demonstrate the disentanglement of skills and knowledge.
> > >
> > > |                | HotpoQA (EM) | Biography (Memorized entities) |
> > > |----------------|--------------|--------------------------------|
> > > | $\mathrm{SFT}^\mathrm{fictitious}$ | 52.50        | 31.21%                         |
> > > | Prereq-Tune    |     11.98         | 12.41%                         |
> > > | $\mathrm{SFT}^\mathrm{real}$       | 10.33        | 10.56%                         |
> > >
> > > [1] Kandpal et al., Large Language Models Struggle to Learn Long-Tail Knowledge.
> > >
> > > [2] Mallen et al., When Not to Trust Language Models: Investigating Effectiveness of Parametric and Non-Parametric Memories.
> > >
> > > [3] Min et al., FActScore: Fine-grained Atomic Evaluation of Factual Precision in Long Form Text Generation.

---

> > > > ### Author Response · Authors · 2024-12-02
> > > >
> > > > Dear reviewer D52y,
> > > >
> > > > We would like to follow up on our pending discussion about the performance of our method on unfamiliar data and its generalizability across different base LLMs. We have added experiments to address your concerns as outlined in our previous responses.
> > > >
> > > > Regarding the generalizability to different base LLMs, we added the experiments on QWEN-14.8B. In addition, we have conducted another experiment to evaluate our method with an even larger LLM, Gemma 2 27B. The following table shows the performance on HotpotQA, where our method continues to outperform the baselines.
> > > >
> > > > |             | Comparison EM | Bridging EM |
> > > > |-------------|---------------|-------------|
> > > > | SFT         | 68.75         | 32.06       |
> > > > | Popular     | 66.51         | 30.53       |
> > > > | Prereq-Tune | **68.94**         | **32.86**       |
> > > >
> > > > As the discussion period will end today, we look forward to your feedback on whether our responses have addressed the concerns.

---

> > > > > ### Comment · Reviewer_D52y · 2024-12-02
> > > > > **Response**
> > > > >
> > > > > I appreciate the substantial efforts the authors put into addressing my concerns and am impressed by the good results. I think this is a good paper now if the authors could include the rebuttal content in the paper. Good luck!

---

### Author Response · Authors · 2024-11-24
**General response to reviewers**

Dear reviewers,

Thank you for your valuable comments! We have updated our paper in response to the concerns raised. The changes we made are highlighted in blue. The major changes include:

- We modify Sections 1 and 3.1 to clarify the motivation of our paper. We additionally include the definition of ‘skill’ and its relation to existing literature in Section 3.1.
- We add an experiment on multihop QA in Sections 4.1 and 4.2. These new results suggest that our method can generalize to more complex tasks involving multiple reasoning steps.
- We add an experiment in Appendix D.3 to evaluate the generalizability of models trained with our method. The results show that the knowledge grounding ability can be generalized across different domains.
- We add an experiment in Appendix D.4 comparing with a baseline that mixes the knowledge and task datasets and trains a single LoRA on them. The results illustrate the importance of the two-step training strategy employed by Prereq-Tune.

 We would really appreciate your feedback on whether our changes have addressed your concerns. Thank you for your time.

---

### Meta-Review · Area_Chair_H875 · 2024-12-21

**Metareview:**

Summary of the paper: This paper introduces PREREQ-TUNE, a fine-tuning strategy designed to enhance the factuality of LLMs while reducing hallucinations. The core idea lies in separating the learning of skills from the acquisition of knowledge through a prerequisite learning stage. This initial stage employs a "knowledge LoRA" to train the model on essential background information, allowing the subsequent "skill LoRA" to concentrate solely on task-specific abilities without being misled by inconsistencies in knowledge. Experiments on biography generation and medical question-answering demonstrate that PREREQ-TUNE significantly outperforms baselines in minimizing hallucinations. The approach also effectively utilizes synthetic data to enhance LLMs' output groundedness and shows resilience against knowledge pollution. Additionally, this paper explores extensions for answer abstention and expressing uncertainty, highlighting the potential for modular LLM designs.

Strengths of the paper:
- Clear Presentation: In general, this paper is well-written and easy to follow, with well-designed figures and tables that facilitate comprehension. However, it would benefit from addressing some minor points (see below).
- Innovative and Effective Approach: The skill LoRA concept for grounding is both innovative and straightforward, supported by clear examples that enhance understanding.
- Compelling Empirical Results: Experiments convincingly demonstrate the effectiveness of skill LoRA, particularly in its limitations when relevant knowledge is missing, addressing the significant issue of model hallucination.
- Scalable Solution: The method's reliance on synthetically generated data allows for scalability, enhancing its practical application and performance over standard fine-tuning techniques.

Weaknesses of the paper:
- Clarification of the motivation (Reviewer D52y): A more thorough explanation of the motivation is needed, particularly regarding the definition of "skills" and its connection to existing literature. This would enhance the reader's understanding of the paper's objectives.
- Generalization Concerns: (Reviewer D52y, G14i, ygrh, Qkzs): There are significant concerns about the generalizability of the proposed method. It remains uncertain whether the approach can be effectively applied to unfamiliar datasets and domains. Additionally, questions arise about its applicability to a broader range of LLMs, particularly those beyond the Llama family or with larger parameters. The method's effectiveness in handling complex reasoning tasks, as opposed to straightforward factual questions, requires further investigation.

Reasons for the decision: After considering the rebuttal, I believe the authors have adequately addressed most of the concerns raised by the reviewers, as outlined in the reviewer discussion box. Although some reviewers did not respond, I took the time to review their comments, and I found that the authors effectively addressed the identified issues. All reviewers agree that this paper should be accepted at ICLR. Upon careful reflection, I believe this paper presents a promising and innovative solution to a key issue in LLMs: improving factuality and reducing hallucinations. Its scalability and potential to engage the broader community make it a valuable contribution that could have an impact.

**Additional Comments On Reviewer Discussion:**

The authors have adequately addressed most of the reviewers' concerns. The paper has been revised accordingly, adding more discussions and experiments as detailed below:

- Clarification of Motivation (Reviewer D52y): The authors have modified Sections 1 and 3.1 to better articulate the motivation behind their work. They also included a definition of 'skill' and its relation to existing literature in Section 3.1.
- Generalization Concerns (Reviewers D52y, G14i, ygrh, Qkzs): The authors conducted an additional experiment demonstrating that their method significantly outperforms SFT, particularly when trained on unfamiliar datasets. Specifically, they evaluated the Qwen 2.5 model with 14.8B parameters on biography generation and multihop QA tasks, achieving the best performance. This indicates its generalizability across different base LLMs. Furthermore, they performed an additional experiment focused on the more complex task of multi-hop reasoning. To assess whether the knowledge grounding enhanced by their approach generalizes across various domains, the authors conducted experiments in one domain and tested the results in another.

---

### Decision · Program_Chairs · 2025-01-22

Accept (Poster)